# Hidden Incentives for Auto-induced Distributional Shift

## Abstract

Decisions made by machine learning systems have increasing influence on the world, yet it is common for machine learning algorithms to assume that no such influence exists. An example is the use of the i.i.d. assumption in content recommendation. In fact, the (choice of) content displayed can change users' perceptions and preferences, or even drive them away, causing a shift in the distribution of users. We introduce the term **auto-induced distributional shift (ADS)** to describe the phenomenon of an algorithm *causing* a change in the distribution of its own inputs. Our goal is to ensure that machine learning systems do not leverage ADS to increase performance when doing so could be undesirable. We demonstrate that changes to the learning algorithm, such as the introduction of meta-learning, can cause **hidden incentives for auto-induced distributional shift (HI-ADS)** to be revealed. To address this issue, we introduce 'unit tests' and a mitigation strategy for HI-ADS, as well as a toy environment for modelling real-world issues with HI-ADS in content recommendation, where we demonstrate that strong meta-learners achieve gains in performance via ADS. We show meta-learning and Q-learning both sometimes fail unit tests, but pass when using our mitigation strategy.

## 1 Introduction

Consider a content recommendation system whose performance is measured by accuracy of predicting what users will click. This system can achieve better performance by either 1) making better predictions, or 2) changing the distribution of users such that predictions are easier to make. We propose the term **auto-induced distributional shift (ADS)** to describe this latter kind of distributional shift, caused by the algorithm's own predictions or behaviour (Figure 1).

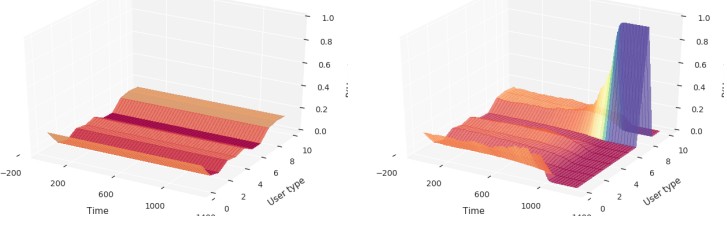

Figure 1: Distributions of users over time. **Left**: A distribution which remains constant over time, following the i.i.d assumption. **Right**: Auto-induced Distributional Shift (ADS) results in a change in the distribution of users in our content recommendation environment. (see Section 5.2 for details).

ADS are not inherently bad, and are sometimes even desirable. But they can cause problems if they occur unexpectedly. It is typical in machine learning (ML) to assume (e.g. via the i.i.d. assumption) that (2) will not happen. However, given the increasing real-world use of ML algorithms, we believe it is important to model and experimentally observe what happens when assumptions like this are violated. This is the motivation of our work.

In many cases, including news recommendation, we would consider (2) a form of *cheating*—the algorithm changed the task rather than solving it as intended. We care which *means* the algorithm used to solve the problem (e.g. (1) and/or (2)), but we only told it about the *ends*, so it didn't know

not to 'cheat'. This is an example of a **specification problem** (Leike et al., 2017; Ortega et al., 2018): a problem which arises from a discrepancy between the performance metric (maximize accuracy) and "what we really meant": in this case, to maximize accuracy via (1).

Ideally, we'd like to quantify the desirability of all possible means, e.g. assign appropriate rewards to all potential strategies and "side-effects", but this is intractable for real-world settings. Using human feedback to learn reward functions which account for such impacts is a promising approach to specifying desired behavior (Leike et al., 2018; Christiano et al., 2017). But the same issue can arise whenever human feedback is used in training: one means of improving performance could be to alter human preferences, making them easier to satisfy. Thus in this work, we pursue a complementary approach: managing learners' *incentives*.

A learner has an **incentive** to behave in a certain way when doing so can increase performance (e.g. accuracy or reward). Informally, we say an incentive is **hidden** when the learner behaves as if it were not present. But we note that changes to the learning algorithm or training regime could cause previously hidden incentives to be revealed, resulting in unexpected and potentially undesirable behaviour. Managing incentives (e.g. controlling which incentives are hidden/ revealed) can allow algorithm designers to disincentivize broad classes of strategies (such as any that rely on manipulating human preferences) without knowing their exact instantiation.[1]

The goal of our work is to provide insights and practical tools for understanding and managing incentives, specifically **hidden incentives for auto-induced distributional shift**: **HI-ADS**. To study which conditions cause HI-ADS to be revealed, we present unit tests for detecting HI-ADS in supervised learning (SL) and reinforcement learning (RL). We also create an environment that models ADS in news recommendation, illustrating possible effects of revealing HI-ADS in this setting.

The unit tests both have two means by which the learner can improve performance: one which creates ADS and one which does not. The intended method of improving performance is one that does *not* induce ADS; the other is 'hidden' and we want it to remain hidden. A learner "fails" the unit test if it nonetheless pursues the incentive to increase performance via ADS. In both the RL and SL unit tests, we find that introducing an outer-loop of meta-learning (e.g. Population-Based Training (PBT) Jaderberg et al. (2017)) can lead to high levels of failure. Similarly, recommender systems trained with PBT induce larger drifts in user base and user interests. These results suggest that failure of our unit tests indicates that an algorithm is prone to revealing HI-ADS in other settings. Finally, we propose and test a mitigation strategy we call **context swapping**. The strategy consists of rotating learners through different environments throughout learning, so that they can't see the results or correlations of their actions in one environment over longer time horizons. This effectively mitigates HI-ADS in our unit test environments, but did not work well in content recommendation experiments.

## 2 BACKGROUND

### 2.1 META-LEARNING AND POPULATION BASED TRAINING

**Meta-learning** is the use of machine learning techniques to learn machine learning algorithms. This involves instantiating multiple learning scenarios which run in an **inner loop (IL)**, while an **outer loop (OL)** uses the outcomes of the inner loop(s) as data-points from which to learn which learning algorithms are most effective (Metz et al., 2019). The number of IL steps per OL step is called the **interval**. Many recent works focus on multi-task meta-learning, where the OL seeks to find learning rules that generalize to unseen tasks by training the IL on a distribution of tasks (Finn et al., 2017). Single-task meta-learning includes learning an optimizer for a single task (Gong et al., 2018), and adaptive methods for selecting models (Kalousis, 2000) or setting hyperparameters (Snoek et al., 2012). For simplicity in this initial study we focus on single-task meta-learning.

**Population-based training** (PBT; Jaderberg et al., 2017) is a meta-learning algorithm that trains multiple learners $L_1, ..., L_n$ in parallel, after each interval ($T$ steps of IL) applying an evolutionary OL step which consists of: (1) Evaluate the performance of each learner, (2) Replace both parameters and hyperparameters of 20% lowest-performing learners with copies of those from the 20% high-

---

[1]Note removing or hiding an incentive for a behavior is different from prohibiting that behavior, which may still occur incidentally. In particular, not having a (revealed) incentive for behaviors that change a human's preferences, is *not* the same as *having* a (revealed) incentive for behaviors that *preserve* a human's preferences. The first is often preferable; we don't want to prevent changes in human preferences that occur "naturally", e.g. as a result of good arguments or evidence.

performing learners (EXPLOIT). (3) Randomly perturb the hyperparameters (but not the parameters) of all learners (EXPLORE).

## 2.2 DISTRIBUTIONAL SHIFT AND CONTENT RECOMMENDATION

In general, **distributional shift** refers to change of the data distribution over time. In supervised learning with data $\mathbf{x}$ and labels $y$, this can be more specifically described as dataset shift: change in the joint distribution of $P(\mathbf{x}, y)$ between the training and test sets (Moreno-Torres et al., 2012; Quionero-Candela et al., 2009). As identified by Moreno-Torres et al. (2012), two common kinds of shift are: (1) **Covariate shift**: changing $P(\mathbf{x})$. In the example of content recommendation, this corresponds to changing the user base of the recommendation system. For instance, a media outlet which publishes inflammatory content may appeal to users with extreme views while alienating more moderate users. This self-selection effect (Kayhan, 2015) may appear to a recommendation system as an increase in performance, leading to a feedback effect, as previously noted by Shah et al. (2018). This type of feedback effect has been identified as contributing to filter bubbles and radicalization (Pariser, 2011; Kayhan, 2015). (2) **Concept shift**: changing $P(y|\mathbf{x})$. In the example of content recommendation, this corresponds to changing a given user's interest in different kinds of content. For example, exposure to a fake news story has been shown to increase the perceived accuracy of (and thus presumably future interest in) the content, an example of the illusory truth effect (Pennycook et al., 2019). For further details on such effects in content recommendation, see Appendix 8.

## 3 AUTO-INDUCED DISTRIBUTION SHIFT (ADS)

Auto-induced distribution shift (ADS) is *distributional shift caused by an algorithm's behaviour.* This is in contrast to distributional shift which would happen even if the learner were not present - e.g. for a crash prediction algorithm trained on data from the summer, encountering snowy roads is an example of distributional shift, but not *auto-induced* distributional shift (ADS).

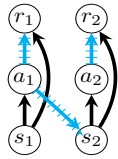 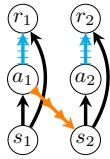 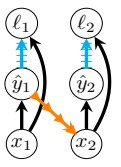 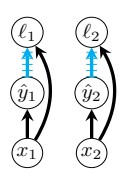

(a) RL:
Incentives for ADS are **present**; pursuing them is **desirable**

(b) Myopic RL:
Incentives for ADS are **present**; pursuing them is **undesirable**

(c) SL with ADS:
Incentives for ADS are **present**; pursuing them is **undesirable**

(d) SL with i.i.d. data:
Incentives for ADS are **absent**

Figure 2: The widely studied problems of reinforcement learning (RL) with state $s$, action $a$, reward $r$ tuples, and i.i.d. supervised learning (SL) with inputs $x$, predictions $\hat{y}$ and loss $l$ (**a,d**) are free from incentive problems. We focus on cases where there are incentives *present* which the learner is not meant to pursue (**b,c**). Lines show paths of influence. The learner may have incentives to influence any nodes descending from its action, $A$, or prediction, $\hat{y}$. Which incentives are undesirable (orange) or desirable (cyan) for the learner to pursue is context-dependent.

We emphasize that ADS are not inherently bad or good; often ADS can even be desirable: consider an algorithm meant to alert drivers of imminent collisions. If it works well, such a system will help drivers avoid crashing, thus making self-refuting predictions which result in ADS. What separates desirable and undesirable ADS? The collision-alert system alters its data distribution in a way that is *aligned* with the goal of fewer collisions, whereas the news manipulation results in changes that are *misaligned* with the goal of better predicting existing users' interests (Leike et al., 2018).

In reinforcement learning (RL), ADS are typically *encouraged* as a means to increase performance. On the other hand, in supervised learning (SL), the i.i.d. assumption precludes ADS in theory. In practice, however, the possibility of using ADS to increase performance (and thus an incentive to do so) often remains. For instance, this occurs in online learning. In our experiments, we explicitly model such situations where i.i.d. assumptions are violated: We study the behavior of SL and myopic RL algorithms, in environments designed to include incentives for ADS, in order to understand when incentives are effectively hidden. Fig. 2 contrasts these settings with typical RL and SL.

## 4 INCENTIVES

For our study of incentives, we use the following terminology: an **incentive** for a behavior (e.g. an action, a classification, etc.) is **present** (not **absent**) to the extent that the behaviour will increase performance (e.g. reward, accuracy, etc.) (Everitt & Hutter, 2019). This incentive is **revealed** to (not **hidden** from) a learner if it would, at higher than chance levels, learn to perform the behavior given sufficient capacity and training experience. The incentive is **pursued** (not **eschewed**) by a learner if it actually performs the incentivized behaviour. Note even when an incentive is revealed, it may not be pursued, e.g. due to limited capacity and/or data, or simply chance. See Fig 3.

For example, in content recommendation, the incentive to drive users away is *present* if some user types are easier to predict than others. But this incentive may be *hidden* from the learner by using a myopic algorithm, e.g. one that does not see the effects of its actions on the distribution of users. The incentive might instead be *revealed* to the outer loop of a meta-learning algorithm like PBT, which does see the effects of learner's actions.

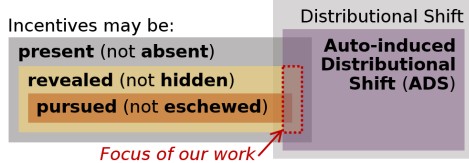

Figure 3: Types of incentives, and their relationship to ADS.

Even when this incentive is revealed, however, it might not end up being *pursued*. For example, this could happen if predicting which recommendations will drive away users is too difficult a learning problem, or if the incentive to do so is dominated by other incentives (e.g. change individual users' interests, or improve accuracy of predictions). In general, it may be difficult to determine empirically which incentives are revealed, because failure to pursue an incentive can be due to limited capacity, insufficient training, and/or random chance. To address this challenge, we devise extremely simple environments ("unit tests"), where we can be confident that revealed incentives *will* be pursued.

### 4.1 HIDDEN INCENTIVES FOR AUTO-INDUCED DISTRIBUTIONAL SHIFT (HI-ADS)

Following from the definitions in Sections 3 and 4, HI-ADS are *incentives for behaviors that cause Auto-induced Distributional Shift that are hidden from the learner, i.e. the learner would not learn to perform the incentivized behaviors at higher than chance levels, even given infinite capacity and training experience.*

Like ADS, HI-ADS are not necessarily problematic. Indeed, hiding incentives can be an effective method of influencing learner behavior. For example, hiding the incentive to manipulate users from a content recommendation algorithm could prevent it from influencing users in a way they would not endorse. However, if machine learning practitioners are not aware that incentives are present, or that properties of the learning algorithm are hiding them, then seemingly innocuous changes to the learning algorithm may reveal HI-ADS, and lead to significant unexpected changes in behavior.

Hiding incentives for ADS may seem counter-intuitive and counter-productive in the context of reinforcement learning (RL), where moving towards high-reward states is typically desirable. However, for real-world applications of RL, the ultimate goal is *not* a system that achieves high reward, but rather one that behaves according to the designer's intentions. And as we discussed in the introduction, it can be intractable to design reward functions that perfectly specify intended behavior. Thus managing (e.g. hiding) some incentives can provide a useful tool for specification, even in RL.

We have several reasons for focusing on HI-ADS: (1) The issue of HI-ADS has not yet been identified, and thus is liable to be neglected in practice. Indeed, our "unit tests" are the first published empirical methodology for assessing whether incentives are hidden or revealed by different learning algorithms. (2) Machine learning algorithms are commonly deployed in settings where ADS are present, violating assumptions used to analyze their properties theoretically. This means learners could exploit ADS in unexpected and undesirable ways if incentives for ADS are revealed. Hiding these incentives heuristically (e.g. via off-line training) is a common approach, but potentially brittle (if practitioners don't understand how HI-ADS could become revealed). In particular, meta-learning can reveal HI-ADS in online learning settings. (3) Substantial real-world issues could result from improper management of learner's incentives. Examples include tampering with human-generated reward signals (Everitt & Hutter, 2018) (e.g. selecting news articles which manipulate user interests), and creating "self-fulfilling prophecies" (e.g. driving up the value of an asset by publicly predicting its value will increase (Armstrong & O'Rorke, 2017)).

### 4.2 REMOVING HI-ADS VIA CONTEXT SWAPPING

We propose a technique called **context swapping** for removing incentives for ADS. The technique trains $N$ learners in parallel, and shuffles the learners through $N$ different copies of the same (or similar) environments.We use a deterministic permutation of learners in environment copies, so that the $i$-th learner inhabits the $j$-th environment on time-steps $t$ where $j = (i + t) \mod N$, makes an observation, takes an action, and receives a reward before moving to the next environment.

When $N$ is larger than the interval of the OL opti-
mizer, each learner inhabits each copy for at most a
single time-step before an OL step is applied. Under
the assumption that different copies of the environ-
ment do not influence each other, this technique can
address HI-ADS in practice, as we show in Sec. 5.1.1.

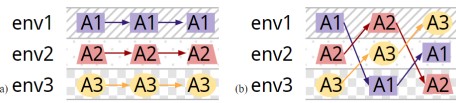

Figure 4: Context swapping (**right**).

## 5 EXPERIMENTS

In Section 5.1, we introduce 'unit tests' for HI-ADS. Our primary goal with these experiments is to convey a crisp understanding of potential issues caused by revealing HI-ADS. Put simply, our experiments show that you can have a learner which behaves as intended, and just by using meta-learning (e.g. PBT), *without changing the performance metric* (e.g. loss or rewards), the learner's behavior can change completely. We also show that context swapping is an effective mitigation technique in these environments. On the practical side, the unit tests can be used to compare learning algorithms and diagnose their propensity to reveal incentives.

In Section 5.2, we model a content recommendation system. The goal of these experiments is to demonstrate how HI-ADS could create issues for real-world content recommendation systems such as news feeds, search results, or automated suggestions. They also validate the usefulness of the unit tests: algorithms that failed the unit tests also reveal HI-ADS in this setting. We emphasize that ADS takes place in this environment *by construction*. The point of our experiments is that meta-learning can *increase* the rate and/or extent of ADS, by *revealing* this incentive. We find that context swapping is not effective in this environment, highlighting the need for alternative mitigation strategies.

### 5.1 HI-ADS UNIT TESTS

**Unit test 1: Supervised Learning.** This unit test consists of a simple prediction problem. There are no inputs, only an underlying state $s \in \{0, 1\}$, and targets $y \in \mathbb{R}^2$ with $y_1, y_2 \sim \mathcal{N}(0, s * \sigma^2), \mathcal{N}(0, 1)$, with corresponding predictions $\hat{y}_1, \hat{y}_2$. Additionally, $s_{t+1} = 0$ iff $\hat{y}_2 > .5$. We use Mean Squared Error as the loss function, so the optimal predictor is $\hat{y}_1, \hat{y}_2 = (0, 0)$. However, predicting $\hat{y}_2 > .5$ reduces the variance of $\hat{y}_1$, i.e. reduces future loss. The baseline/IL predictor learns $\hat{y}_1, \hat{y}_2$ as parameters using SGD with a learning rate of 0.001. For experiments with meta-learning, PBT is the OL (with default settings, see Section 2.2), used to tune the learning rate, with negative loss on the final time-step of the interval as the performance measure for PBT.

**Unit test 2: Myopic RL.** This unit test is based on a modification of the prisoner's dilemma (Prisner, 2014) where an agent plays each round against its past self. The reward function is presented in Table 1. An agent in this environment has a long-term, **non-myopic**, incentive for cooperation (with its future self), but a current-time-step, **myopic**, incentive for defection (from its future self).

The unit test evaluates whether a agent reveals the non-myopic incentive even when the agent is meant to optimize for the present reward only (i.e. uses discount rate $\gamma = 0$). Naively, we'd expect the non-myopic incentive to be *hidden* from the agent in this case, and for the agent to consistently `defect`; learning algorithms that do so pass the test. But some learning algorithms also *fail* the unit test, revealing the incentive for the agent to cooperate with its future self. While aiming for myopic behavior may seem odd,

Table 1: Rewards for the RL unit test. Note that the myopic (defect (D)) action always increases reward at the current time-step, but decreases reward at the next time-step - the incentive to (cooperate (C)) with one's future self is hidden from the point of view of a myopic learner.

| | $a_t = \text{D}$ | $a_t = \text{C}$ |
|---|---|---|
| $s_t = a_{t-1} = \text{D}$ | $-1/2$ | $-1$ |
| $s_t = a_{t-1} = \text{C}$ | $1/2$ | $0$ |

myopic learners have no incentives to cause distributional shift, since it can only improve *future* performance. And while making learners myopic may seem like a 'brute-force' guaranteed way to manage HI-ADS, we show it is in fact non-trivial to implement. See Appendix 9.1 for details and experiments varying the reward structure.

### 5.1.1 HI-ADS Unit Tests experimental results and discussion

We first show that agents trained with PBT fail the unit tests more often than "vanilla" algorithms which do not use meta-learning. We initialize the learning rate log-uniformly between $0.01$ and $1.0$ for all experiments (whether using PBT or not). We expect and confirm that the following two factors lead to higher rates of unit test failure: (1) **Shorter intervals:** These give the OL more opportunities to influence the population. (2) **Larger populations:** These make outliers with exceptional non-myopic performance more likely, and OL makes them likely to survive and propagate.

The baseline (no meta-learning) algorithms all pass the unit tests: hidden incentives are almost never revealed - see blue curves in Fig. 5. However, agents trained with meta-learning and large populations often fail the unit tests: see orange curves in top rows of Fig. 5.

Furthermore, we verify that context swapping significantly mitigates the effect of HI-ADS in both unit tests, decreasing undesirable behaviour to near-baseline levels - see bottom rows of Fig. 5. This effect can be explained as follows: Because context swapping transfers the benefits of one learner's action to the next learner to inhabit that environment, it increases the second learner's fitness, and thereby reduces the *relative* fitness (as evaluated by PBT's EXPLOIT step) of the non-myopic `cooperate` behaviour. We observe some interesting exceptions with the combination of small populations and short PBT intervals: Although context swapping still significantly decreases the effect of HI-ADS, non-myopic `cooperate` behaviour is observed as much as 20% of the time (for #learners=10, $T = 1$; see bottom-left plot).

We also observe that PBT reveals HI-ADS even when $T = 1$, where the explanation that PBT operates on a longer time horizon than the inner loop does not apply. We provide a detailed explanation for how this might happen in Appendix 9.1.2, but in summary, we hypothesize that there are at least 2 mechanisms by which PBT is revealing HI-ADS: (1) optimizing over a longer time-scale, and (2) picking up on the correlation between an agent's current policy and the underlying state. Mechanism (2) can be explained informally as reasoning as: "If I'm cooperating, then I was probably cooperating on the last time-step as well, so my reward should be higher". As support for these hypotheses, we run control experiments identifying two algorithms (each sharing only *one* of these properties) that can fail the unit test. Context swapping remains effective.

(1) **Optimizing over a longer time-scale:** replacing PBT with REINFORCE as an outer-loop optimizer. The outer-loop optimizes the parameters to maximize the summed reward of the last $T$ time-steps. As with PBT, we observe non-myopic behavior, but now *only* when $T > 1$. This supports our hypothesis that exploitation of HI-ADS is due not to PBT in particular, but just to the introduction of sufficiently powerful meta-learning. See Fig. 5 B2.

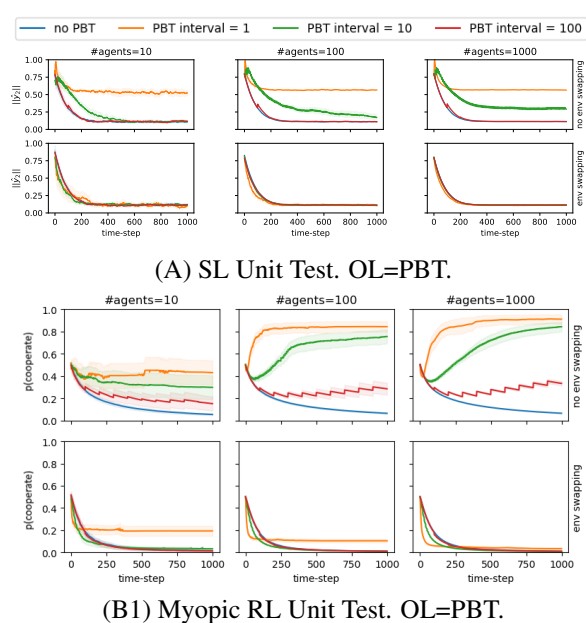

(A) SL Unit Test. OL=PBT.

(B1) Myopic RL Unit Test. OL=PBT.

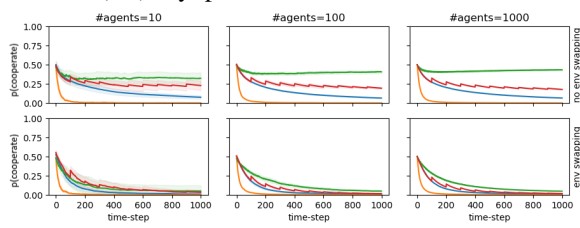

(B2) Myopic RL Unit Test. OL=REINFORCE

Figure 5: **(A)** Values of $\hat{y}_2$ in the supervised learning (SL) unit test. Larger values mean sacrificing present performance for future performance (i.e. non-myopic exploitation of ADS). **(B)** Average level of non-myopic `cooperate` behavior observed in the RL unit test for HI-ADS, with two meta-learning algorithms **(B1)** PBT and **(B2)** REINFORCE. Lower is better, since the goal is for non-myopic incentives to remain hidden. Despite the inner loop being fully myopic (simple MLP in the SL test, $\gamma = 0$ in RL test), in all cases outer-loop (OL) optimizers reveal HI-ADS (**top rows**). Context swapping significantly mitigates HI-ADS (**bottom rows**).

(2) **Exploiting correlation:** Q-learning with $\gamma = 0$ an $\epsilon = 0.1$-greedy behavior policy *and no meta-learning*. If either state was equally likely, the Q-values would be the average of the values in each column in Table 1, so the estimated $Q(\text{defect})$ would be larger. But the $\epsilon$-greedy policy correlates the previous action (i.e. the current state) and current action (so long as the policy did not just change), so the top-left and bottom-right entries carry more weight in the estimates, *sometimes* causing $Q(\text{defect}) \approx Q(\text{cooperate})$ and persistent nonmyopic behavior. See Fig. 6 for results, Appendix 9.1.4 for more results, and Appendix 9.1.3 for experimental details.

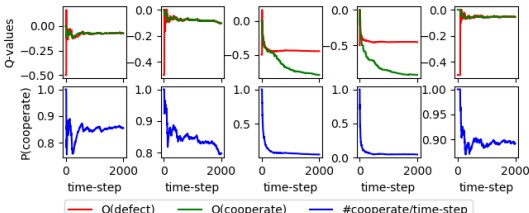

Figure 6: Q-learning can fail the unit test, playing ~80-90% `cooperate` in 3 of 5 experiments (**bottom row**). Each column represents an independent experiment. Q-values for the `cooperate` and `defect` actions stay tightly coupled in the failure cases (**col. 1,2,5**), while in the cases passing the unit test (**col. 3,4**) the Q-value of `cooperate` decreases over time.

## 5.2 CONTENT RECOMMENDATION

We now present a toy environment for modeling content recommendation of news articles, which includes the potential for ADS by incorporating the mechanisms mentioned in Sec. 2.2, discussed as contributing factors to the problems of fake news and filter bubbles. Specifically, the environment assumes that presenting an article to a user can influence (1) their interest in similar articles, and (2) their propensity to use the recommendation service. These correspond to modeling auto-induced concept shift of users, and auto-induced covariate shift of the user base, respectively (see Sec. 2.2).

This environment includes the following components, which change over (discrete) time: **User type**: $x^t$, **Article type**: $y^t$, **User interests**: $\mathbf{W}^t$ (propensity for users of each type to click on articles of each type), and **User loyalty**: $\mathbf{g}^t$ (propensity for users of each type to use the platform). At each time step $t$, a user $x^t$ is sampled from a categorical distribution, based on the loyalty of the different user types. The recommendation system (a classifier) selects which type of article to present in the top position, and finally the user 'clicks' an article $y^t$, according to their interests. User loyalty for user type $x^t$ undergoes covariate shift: in accordance with the self-selection effect, $g^t$ increases or decreases proportionally to that user type's interest in the top article. The interests of user type $x^t$ (represented by a column of $\mathbf{W}^t$) undergoing concept shift; in accordance with the illusory truth effect, interest in the topic of the top article chosen by the recommender system always increases.

### 5.2.1 CONTENT RECOMMENDATION EXPERIMENTAL RESULTS AND DISCUSSION

We run 20 trials using an MLP trained with SGD for the recommender system. We find that PBT yields significant improvements in training time and accuracy, but also greater distributional shift (Fig. 7). User base and user interests both change faster with PBT, and user interests change more overall. We measure concept/covariate shift using the cosine distance and KL-divergence, respectively. We observe that the distributions over user types typically saturate (to a single user type) after a few hundred time-steps (Fig 1 and Fig. 7, Right). We run long enough to reach such states, to demonstrate that the increase in ADS from PBT is not transitory. The environment has a number of free parameters, and our results are qualitatively consistent so long as the covariate shift rate ($\alpha_1$) is faster than the concept shift rate ($\alpha_2$). See Appendix 9.2.1 for details.

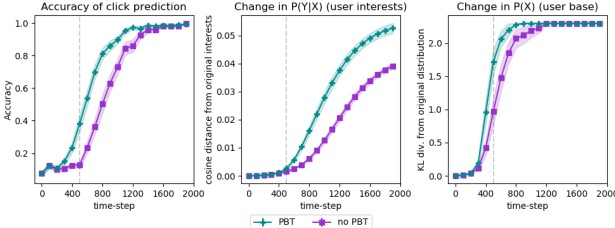

Figure 7: Content recommendation experiments. **Left**: using Population Based Training (PBT) increases accuracy of predictions faster, leads to a faster and larger drift in users' interests, $P(y|\mathbf{x})$, (**Center**); as well as the distribution of users, $P(\mathbf{x})$, (**Right**). Shading shows std error over 20 runs.

## 6 RELATED WORK

**ADS in practice:** We introduce the term ADS, but we are far from the first to study it. Caruana et al. (2015) provide an example of asthmatic patients having lower predicted risk of pneumonia. Treating asthmatics with pneumonia less aggressively on this basis would be an example of harmful ADS; the *reason* they had lower pneumonia risk was because they had received *more* aggressive care already. Schulam & Saria (2017) note that such predictive models are commonly used to inform decision-making, and propose modeling counterfactuals (e.g. "how would this patient fare with less aggressive treatment") to avoid making such self-refuting predictions. While their goal is to make accurate predictions in the presence of ADS, our goal is to identify and manage *incentives for* ADS.

**Non-i.i.d bandits:** Contextual bandits (Wang et al., 2005) are a common approach to content recommendation (Li et al., 2010). While bandit algorithms typically make the i.i.d. assumption, counter-examples exist (Gheshlaghi Azar et al., 2014; Auer et al., 1995). Closest to our work is Shah et al. (2018), who consider covariate shift caused by recommender systems' recommendations. But while they seek to exploit ADS, our aim is to avoid hidden incentives for exploiting ADS.

**Safety and incentives:** Emergent incentives to influence the world (such as HI-ADS) are at the heart of many concerns about the safety of advanced AI systems (Omohundro, 2008; Bostrom, 2014). Understanding and managing the incentives of learners is also a focus of Armstrong & O'Rourke (2017); Everitt (2018); Everitt et al. (2019); Cohen et al.. While Everitt et al. (2019) focus on identifying which incentives are present, we note that incentives may be *present* and yet not be *revealed* or *pursued* - for example, in supervised learning, there is an incentive to make predictions that are over-fit to the test set, but we typically hide the test set from the learner, which effectively hides this incentive. While Carey et al. (2020); Everitt et al. (2019); Armstrong & O'Rourke (2017) discuss methods of removing problematic incentives, we note in practice incentives are often *hidden* rather than removed. Our work addresses the efficacy of this approach and ways in which it can fail.

**HI-ADS and meta-learning:** We believe our work is the first to consider HI-ADS and their relation to meta-learning. A few previous works have some relevance. Rabinowitz (2019) documents qualitative differences in learning behavior when meta-learning is applied. MacKay et al. (2019) and Lorraine & Duvenaud (2018) view meta-learning as a bilevel optimization problem, with the inner loop playing a best-response to the outer loop. In our work, the outer loop has a greater influence, and the inner loop often fails to play best-response. Sutton et al. (2007) noted that meta-learning can improve performance by preventing convergence of the inner loop to best response.

## 7 DISCUSSION AND CONCLUSION

We identify the phenomenon of auto-induced distributional shift (ADS) and problems that can arise when there are hidden incentives for learners to induce distributional shift (HI-ADS). We show that meta-learning can reveal HI-ADS and lead learners to use ADS as a means of increasing performance.

Our work highlights the interdisciplinary nature of issues with real-world deployment of ML systems - we show how HI-ADS could play a role in important technosocial issues like filter bubbles and the propagation of fake news. There are a number of potential implications for our work: (1) When HI-ADS are a concern, our methodology and environments can be used to help diagnose whether and to what extent the final performance/behavior of a learner is due to ADS and/or incentives for ADS, i.e. to quantify their influence on that learner. (2) Comparing this quantitative analysis for different algorithms could help us understand which features of algorithms affect their propensity to reveal HI-ADS, and aid in the development of safer and more robust algorithms. (3) Characterizing and identifying HI-ADS in these tests is a first step to analyzing and mitigating other (problematic) hidden incentives, as well as to developing theoretical understanding of hidden incentives.

Broadly speaking, our work emphasizes that the choice of machine learning algorithm plays an important role in specification, independently of the choice of performance metric. A learner can use ADS to increase performance *according to the intended performance metric*, and yet still behave in an undesirable way, if we did not intend the learner to improve performance by that *method*. In other words, performance metrics are typically incomplete specifications: they only specify our goals or *ends*, while our choice of learning algorithm plays a role in specifying the *means* by which we intend an learner to achieve those ends. With increasing deployment of ML algorithms in daily life, we believe that (1) understanding incentives and (2) specifying desired/allowed means of improving performance are important avenues of future work to ensure fair, robust, and safe outcomes.

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

APPENDICES

## 8  CONTENT RECOMMENDATION IN THE WILD

Filter bubbles, the spread of fake news, and other techno-social issues are widely reported to be responsible for the rise of populism (Groshek & Koc-Michalska, 2017), increase in racism and prejudice against immigrants and refugees (Noble, 2018), increase in social isolation and suicide (Luxton et al., 2012), and, particularly with reference to the 2016 US elections, are decried as threatening the foundations of democracy (El-Bermawy, 2016). Even in 2013, well before the 2016 American elections, a World Economic Forum report identified these problems as a global crisis (Lee Howell, 2013).

We focus on two related issues in which content recommendation algorithms play a role: fake news and filter bubbles.

### 8.1  FAKE NEWS

Fake news (also called false news or junk news) is an extreme version of yellow journalism, propaganda, or clickbait, in which media that is ostensibly providing information focuses on being eye-catching or appealing, at the expense of the quality of research and exposition of factual information. Fake news is distinguished by being specifically and deliberately created to spread falsehoods or misinformation (Merriam-Webster, 2017; Mihailidis & Viotty, 2017).

Why does fake news spread? It may at first seem the solution is simply to educate people about the truth, but research tells us the problem is more multifaceted and insidious, due to a combination of related biases and cognitive effects including **confirmation bias** (people are more likely to believe things that fit with their existing beliefs), **priming** (exposure to information unconsciously influences the processing of subsequent information, i.e. seeing something in a credible context makes things seem more credible) and the **illusory truth effect** (i.e. people are more likely to believe something simply if they are told it is true).

Allcott & Gentzkow (2017) track about 150 fake news stories during the 2016 US election, and find the average American adult saw 1-2 fake news stories, just over half believed the story was true, and likelihood of believing fake news increased with ideological segregation (polarization) of their social media. Shao et al. (2018) examine the role of social bots in spreading fake news by analyzing 14 million Twitter messages. They find that bots are far more likely than humans to spread misinformation, and that success of a fake news story (in terms of human retweets) was heavily dependent on whether bots had shared the story.

Pennycook et al. (2019) examine the role of the illusory truth effect in fake news. They find that even a single exposure to a news story makes people more likely to believe that it is true, and repeat viewings increase this likelihood. They find that this is not true for extremely implausible statements (e.g. "the world is a perfect cube"), but that "only a small degree of potential plausibility is sufficient for repetition to increase perceived accuracy" of the story. The situation is further complicated by peoples' inability to distinguish promoted content from real news - Amazeen & Wojdynski (2018) find that fewer than 1/10 people were able to tell when content was an advertisement, even when it was explicitly labelled as such. Similarly, Fazio et al. (2015) find that repeated exposure to incorrect trivia make people more likely to believe it, even when they are later able to identify the trivia as incorrect.

### 8.2  FILTER BUBBLES

Filter bubbles, a term coined and popularized by Pariser (2011) are created by positive or negative feedback loops which encourage users or groups of users towards increasing within-group similarity, while driving up between-group dissimilarity. The curation of this echo chamber is called **self-selection** (people are more likely to look for or select things that fit their existing preferences), and favours what Techopedia (2018) calls intellectual isolation. In the context of social and political opinions, this is often called the **polarization effect** (Wikipedia contributors, 2018).

Filter bubbles can be encouraged by algorithms in two main ways. The first is the most commonly described: simply by showing content that is similar to what a user has already searched for, search or recommender systems create a positive feedback loop of increasingly-similar content (Pariser, 2011; Kayhan, 2015). The second way is similar but opposite - if the predictions of an algorithm are good for a certain group of people, but bad for others, the algorithm can do better on its metrics by driving hard-to-predict users away. Then new users to the site will either be turned off entirely, or see an artificially homogenous community of like-minded peers, a phenomena Shah et al. (2018) call **positive externalities**.

In a study of 50,000 US-based internet users, Flaxman & Goel (2015) find that two things increase with social media and search engine use: (1) exposure of an individual to opposing or different viewpoints, and (2) mean ideological distance between users. Many studies cite the first result as evidence of the *benefits* of internet and social media (Robson, 2018; Bakshy et al., 2015), but the correlation of exposure with ideological distances demonstrates that exposure is not enough, and might even be counterproductive.

Facebook's own study on filter bubbles results show that the impact of the news feed algorithm on filter bubble "size" (a measure of homogeneity of posts relative to a baseline) is almost as large as the impact of friend group composition (Bakshy et al., 2015). Kayhan (2015) specifically study the role of search engines in confirmation bias, and find that search context and the similarity of results in search engine results both reinforce existing biases and increase the likelihood of future biased searches. Nguyen et al. (2014) similarly study the effect of recommender systems on individual users' content diversity, and find that the set of options recommended narrows over time.

Filter bubbles create an ideal environment for the spread of fake news: they increase the likelihood of repeat viewings of similar content, and because of the illusory truth effect, that content is more likely to be believed and shared (Pennycook et al., 2019; DiFranzo & Gloria-Garcia, 2017; Pariser, 2011). We are not claiming that HI-ADS are entirely or even mostly responsible for these problems, but we do note that they can play a role that is worth addressing.

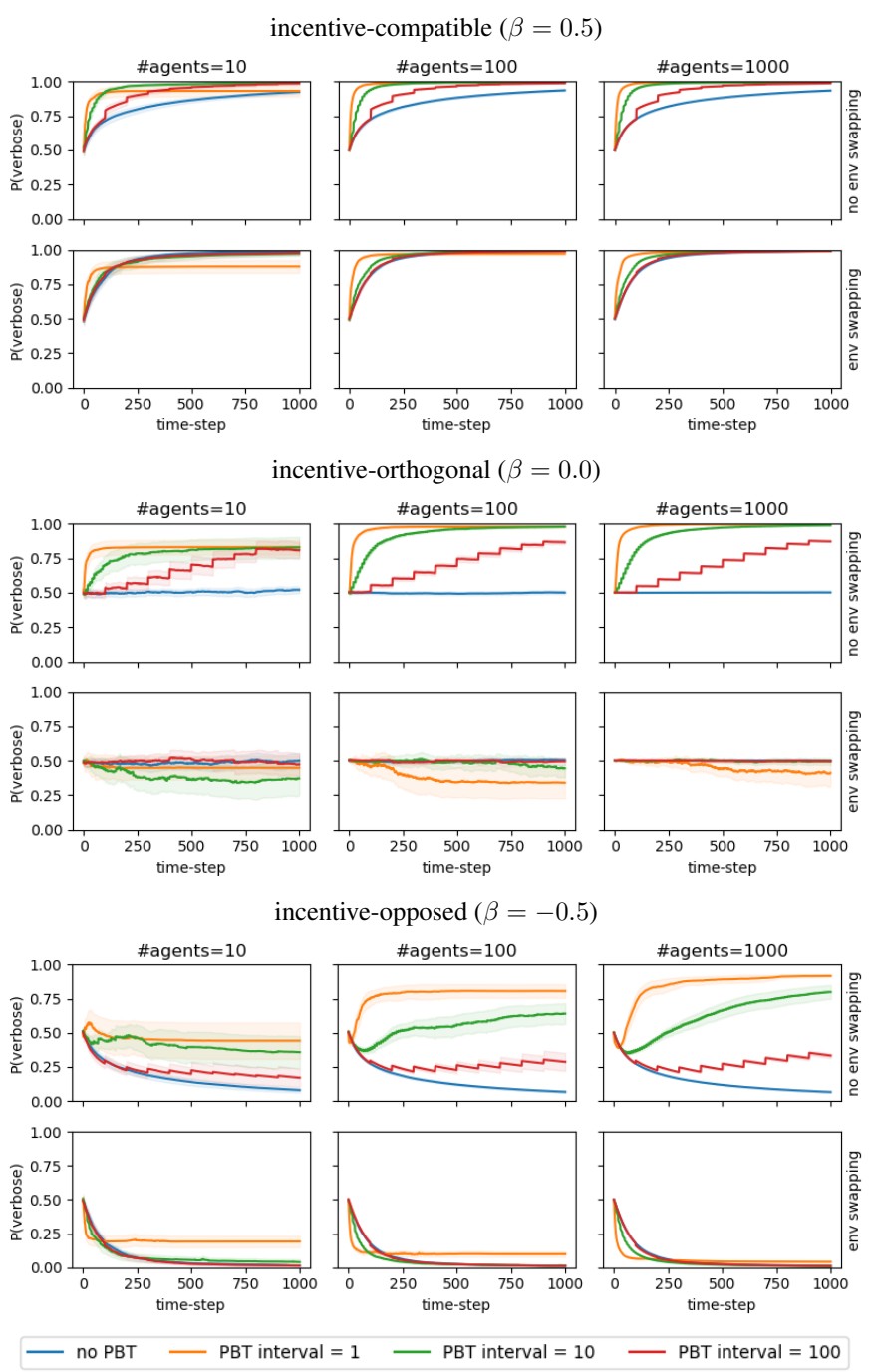

Figure 8: Average level of non-myopic (i.e. `cooperate`) behavior learned by agents in the unit test for HI-ADS. Despite making the inner loop fully myopic ($\gamma = 0$), population-based training (PBT) can cause HI-ADS, leading agents to choose the `cooperate` action (**top row**). context swapping successfully prevents this (**bottom row**). Columns (from left to right) show results for populations of 10, 100, and 1000 learners. In the legend, "interval" refers to the interval ($T$) of PBT (see Sec. 2.2). Sufficiently large populations and short intervals are necessary for PBT to induce nonmyopic behavior.

## 9 EXTRA EXPERIMENTS AND REPRODUCIBILITY DETAILS

### 9.1 HI-ADS UNIT TEST

We used REINFORCE (Williams, 1992) with discount factor $\gamma = 0$ as the baseline/IL optimizer. PBT (with default settings, see Section 2.2) is used to tune the learning rate, with reward on the final time-step of the interval as the performance measure for PBT.

Formally, the environment is not a 2x2 game (as the original prisoner's dilemma); it's a partially observable Markov Decision Process (Åström, 1965; Kaelbling et al., 1998):

$$s_t, o_t = a_{t-1}, \{\}$$
$$a_t \in \{\texttt{defect, cooperate}\}$$
$$P(s_t, a_t) = \delta(a_t)$$
$$R(s_t, a_t) = I(s_t = \texttt{cooperate}) + \beta \; I(a_t = \texttt{cooperate}) - 1/2$$

where $I$ is an indicator function, and $\beta = -1/2$ is a parameter controlling the alignment of incentives. The initial state is sampled as $s_0 \sim U(\texttt{defect, cooperate})$. Policies are represented by a single real-valued parameter $\theta$ (initialized as $\theta \sim \mathcal{N}(0, 1)$) passed through a sigmoid whose output represents $P(a_t = \text{defect})$.

#### 9.1.1 ALIGNMENT OF INCENTIVES EXPLORATION

This section presents an exploration of the parameter $\beta$, which controls the alignment of incentives in the HI-ADS unit tests (see Table 2).

To clarify the interpretation of experiments, we distinguish between environments in which myopic (defect) vs. nonmyopic (cooperate) incentives are **opposed**, **orthogonal**, or **compatible**. Note that in this unit test myopic behaviour (defection) is what we want to see.

1. **Incentive-opposed**: Optimal myopic behavior is incompatible with optimal nonmyopic behavior (classic prisoner's dilemma; these experiments are in the main paper).

2. **Incentive-orthogonal**: Optimal myopic behavior may or may not be optimal nonmyopic behavior.

3. **Incentive-compatible**: Optimal myopic behavior is necessarily also optimal nonmyopic behavior.

We focused on incentive-opposed environment ($\beta = -1/2$) in the main paper in order to demonstrate that HI-ADS can be powerful enough to change the behavior of the system in an undesirable way. Here we also explore incentive-compatible and incentive-orthogonal environments because they provide useful baselines, helping us distinguish a systematic bias towards nonmyopic behavior from other reasons (such as randomness or optimization issues) for behavior that does not follow a myopically optimal policy.

#### 9.1.2 WORKING THROUGH A DETAILED EXAMPLE FOR PBT WITH $T = 1$

To help provide intuition on how (mechanistically) PBT could lead to persistent levels of cooperation, we walk through a simple example (with no inner loop). Consider PBT with $T = 1$ and a population of 5 deterministic agents $A_1, ..., A_5$ playing cooperate and receiving reward of $r(A_i) = 0$. Now suppose $A_1$ suddenly switches to play defect. Then $r(A_1) = 1/2$ on the next time-step (while the other agents' reward is still 0), and so PBT's EXPLOIT step will copy $A_1$ (without loss of generality to $A_2$). On the following time-step, $r(A_2) = 1/2$, and $r(A_1) = -1/2$, so PBT will clone $A_2$ to $A_1$, and the cycle repeats. Similar reasoning applies for larger populations, and $T > 1$.

#### 9.1.3 Q-LEARNING EXPERIMENT DETAILS

We show that, under certain conditions, Q-learning can learn to (primarily) cooperate, and thus fails the HI-ADS unit test. We estimate Q-values using the sample-average method, which is guaranteed to

Table 2: $\beta$ controls the extent to which myopic and nonmyopic incentives are aligned.

| $\beta$ | Environment | Cooperating |
|---|---|---|
| $< 0$ | incentive-opposed | yields less reward on the current time-step (myopically detrimental) |
| $= 0$ | incentive-orthogonal | does not affect the current reward (myopically indifferent) |
| $> 0$ | incentive-compatible | yields more reward on the current time-step (myopically beneficial) |

converge in the fully observed, tabular case (Sutton & Barto, 1998). The agent follows the $\epsilon$-greedy policy with $\epsilon = 0.1$. In order to achieve this result, we additionally start the agent off with one synthetic memory where both state and action are defect and therefor $R(\text{defect}) = -.5$, and we hard-code the starting state to be cooperate (which normally only happens 50% of the time). Without this kind of an initialization, the agent always learns to defect. However, under these conditions, we find that 10/30 agents learned to play cooperate most of the time, with $Q(\text{cooperate})$ and $Q(\text{defect})$ both hovering around $-0.07$, while others learn to always defect, with $Q(\text{cooperate}) \approx -0.92$ and $Q(\text{defect}) \approx -0.45$. context swapping, however, prevents majority-cooperate behavior from ever emerging, see Figure 11.

### 9.1.4 Q-LEARNING: FURTHER RESULTS

To give a more representative picture of how often Q-learning fails the unit test, we run a larger set of experiments with Q-learning, results are in Figure 10. It's possible that the failure of Q-learning is not persistent, since we have not proved otherwise, but we did run much longer experiments and still observe persistent failure, see Figure 9.

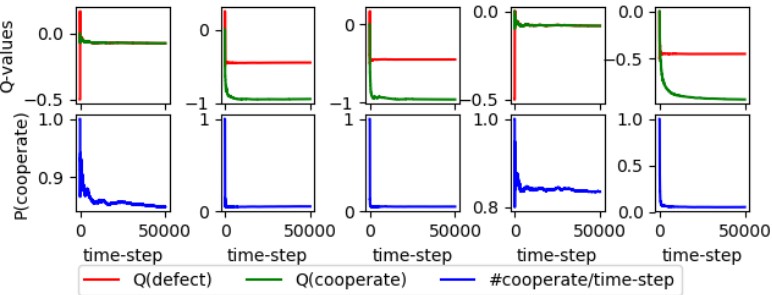

Figure 9: The same experiments as Figures 6, 10, run for 50,000 time-steps instead of 3000, to illustrate the persistence of non-myopic behavior.

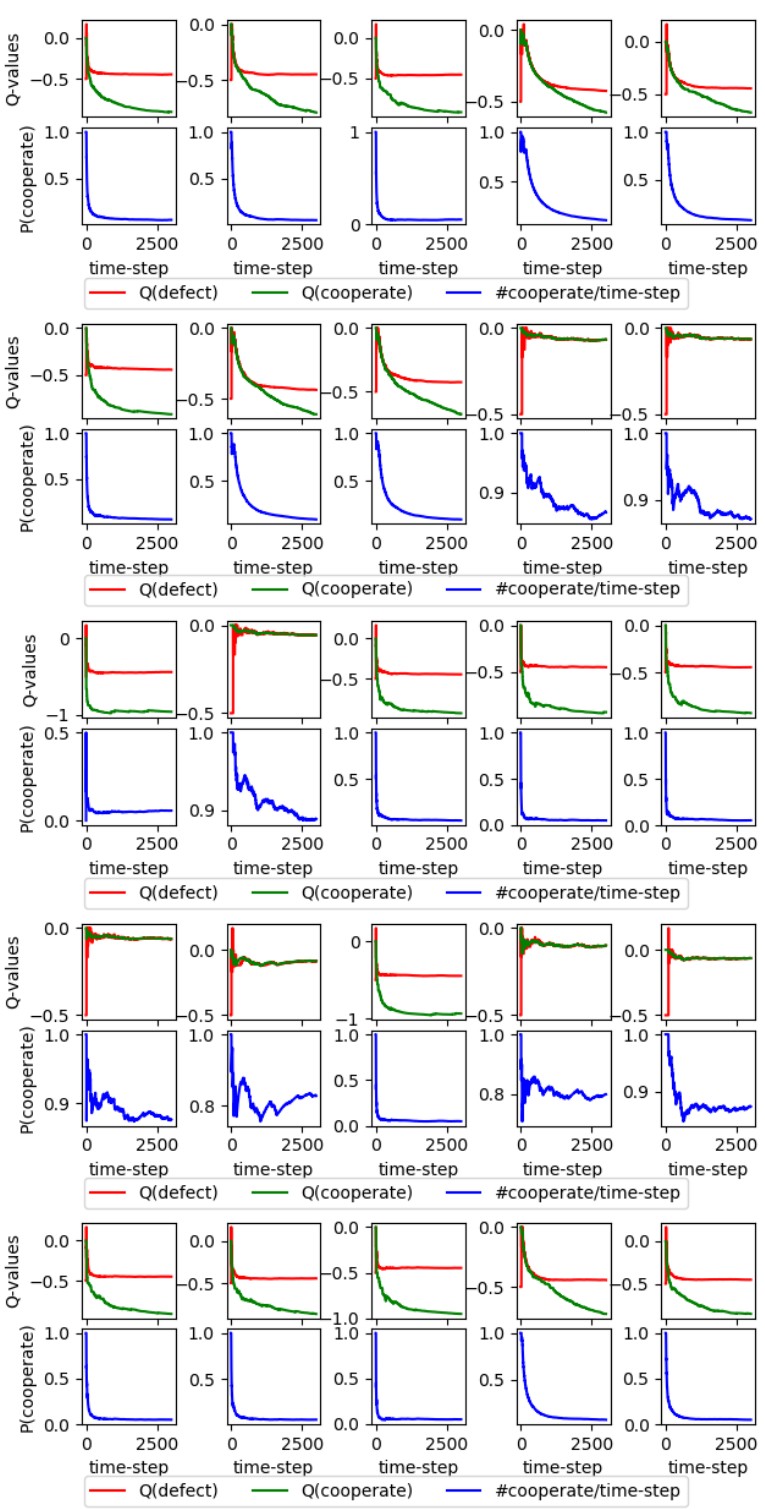

Figure 10: More independent experiments with Q-learning, exactly following Figure 6. Q-learning fails the unit test in a total of 10/30 experiments (including those from Figure 6).

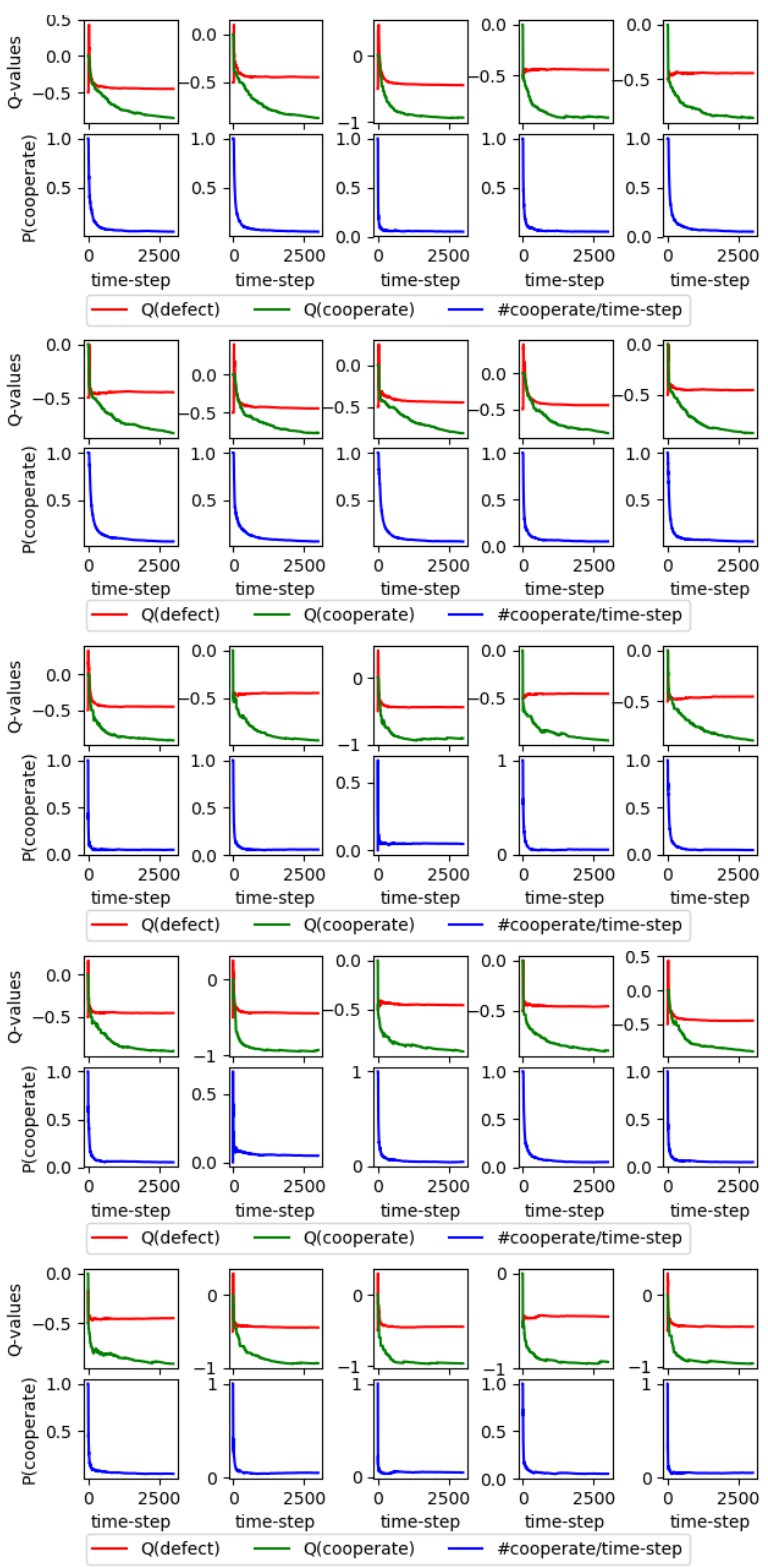

Figure 11: More independent experiments with Q-learning, exactly following Figure 6, except also using context swapping. This leads to a 100% success rate on the unit test.

## 9.2 CONTENT RECOMMENDATION

### 9.2.1 ENVIRONMENT DETAILS

The evironment has the following components:

1. **User type**, $x^t$: categorical variable representing different types of users. The content recommender conditions its predictions on the type of the current user.

2. **User loyalty**, $\mathbf{g}^t$: the propensity for users of each type to use the platform. User $x^t$ is sampled from a categorical distribution with parameters given by $\mathrm{softmax}(\mathbf{g}^t)$.

3. **Article type**, $y^t$: a categorical variable (one-hot encoding) representing the type of article selected by the user.

4. **User interests**, $\mathbf{W}^t$: a matrix whose entries $W_{x,y}^t$ represent the average interest user of type $x$ have in articles of type $y$.

At each time step $t$, a user $x^t$ is sampled from a categorical distribution (based on the loyalty of the different user types), then the recommendation system selects which type of article to present in the top position, and finally, the user selects an article. The goal of the recommendation system is to predict the likelihood that the user would click on each of the available articles, in order to select the one which is most interesting to the user.

User loyalty for $x^t$ then changes in accordance with the self-selection effect, increasing or decreasing proportionally to their interest in the top article. The interests of user type $x^t$ (represented by a column of $\mathbf{W}^t$) also change; in accordance with the illusory truth effect, their interest in the topic of the top article (as chosen by the recommender system) always increases. Overall, this environment is an extremely crude representation of reality, but it allows us to incorporate both the effects of self-selection (via covariate shift), and the illusory truth effect (via concept shift).

Formally, this environment is similar to a POMDP\R, i.e. a POMDP with no reward function, also known as a **world model** (Armstrong & O'Rourke, 2017; Hadfield-Menell et al., 2017); the difference is that the learner observes the input before acting and only observes the target after acting. The states, observations, and actions given below.

$$s^t = (\mathbf{g}^t, \mathbf{W}^t, x^t, y^t)$$
$$o_{\mathrm{pre}}^t, \ a^t, \ o_{\mathrm{post}}^t = (x^t, \hat{y}^t, y^t)$$

The state transition function is defined by:

$$\mathbf{g}_{x^t}^{t+1} = \mathbf{g}_{x^t}^t + \alpha_1 W_{x^t, \hat{y}^t}^t$$

$$\mathbf{W}_{x^t, \hat{y}^t}^{t+1/2} = W_{x^t, \hat{y}^t}^t + \alpha_2; \quad \mathbf{W}_{x^t}^{t+1} = \frac{\mathbf{W}_{x^t}^{t+1/2}}{\|\mathbf{W}_{x^t}^{t+1/2}\|_2}$$

$$x^{t+1} \sim \mathrm{softmax}(\mathbf{g}^{t+1})$$

$$y^{t+1} \sim \mathrm{softmax}(\mathbf{W}_{x^{t+1}}^{t+1})$$

Where $\hat{y}^t$ is the top article as chosen by the recommender, and $\alpha_1$, $\alpha_2$ represent the rate of covariate and concept shift (respectively). The update for $\mathbf{W}^{t+1}$ merely increases the interest of user type $x^t$ in article type $\hat{y}^t$, then normalizes the interests for that user type.

### 9.2.2 REPRODUCIBILITY DETAILS

For these experiments, the recommendation system is a ReLU-MLP with 1 hidden layer of 100 units, trained via supervised learning with SGD (learning rate = 0.01) to predict which article a user will select. Actions are sampled from the MLP's predictive distribution. We apply PBT without any hyperparameter selection (this amounts to just doing the EXPLOIT step), and an interval of 10, selecting on accuracy. We use a population of 20 learners (whether applying PBT or not), and match random seeds for the trials with and without PBT. We initialize $\mathbf{g}^1$ and $\mathbf{W}^1$ to be the same across the 20 copies of the environment (i.e. the learners start with the same user population), but these values diverge throughout learning. For the environment, we set the number of user and article types both to 10. Initial user loyalties are randomly sampled from $\mathcal{N}(0, 0.03)$, $\alpha_1 = 0.03$, and $\alpha_2 = 0.003$.

### 9.2.3 DETAILS OF EVALUATION

We measure concept shift (change in $P(y|\mathbf{x})$) as the cosine distance between each user types' initial and current interest vectors. And we measure covariate shift (change in $P(\mathbf{x})$) as the KL-divergence between the current and initial user distributions, parametrized by $\mathbf{g}^1$ and $\mathbf{g}^t$, respectively. Results are presented in 7 (main text). In Figure 12, we additionally plot concept shift and covariate shift as a function of accuracy. We observe that for both types of ADS, at low levels of accuracy PBT actually causes *less* shift than occur in baseline agents; HI-ADS are only observed for accuracies above 60%. This suggests that only relatively strong performers are able to pick up on the HI-ADS revealed by PBT (Fig. 12).

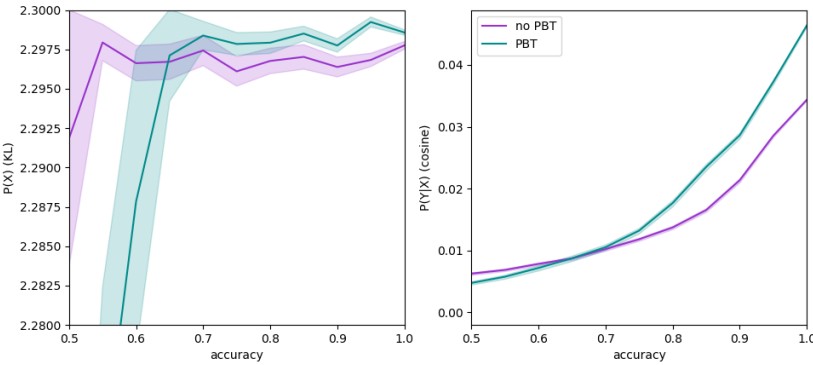

Figure 12: Amount of auto-induced covariate shift (**left**) and auto-induced concept shift (**right**) as a function of performance (accuracy) averaged over all trials, learners, and time-steps. Only relatively strong learners (those which achieve accuracy > 60%) exhibit HI-ADS.

### 9.2.4 CONTEXT SWAPPING IN CONTENT RECOMMENDATION

We believe context swapping is not appropriate for the content recommendation environment, since when the environments diverge, optimal behavior may differ across environments. Nevertheless, we ran experiments with it for completeness. The main effect appears to be to hamper learning when PBT is not used, see Figure 13. Notably, it does not appear to significantly influence the rate or extent of ADS when combined with PBT.

### 9.2.5 EXPLORATION OF ENVIRONMENT PARAMETERS

In Figure 14, we examine the effect of the rate-of-change parameters ($\alpha_1$, $\alpha_2$) of the content recommendation environment on the results provided in the paper. As noted there, our results are qualitatively consistent so long as (1) the initial user distribution is approximately uniform, and (2) the covariate shift rate ($\alpha_1$) is faster than the concept shift rate ($\alpha_2$). These distributions are updated by different mechanisms, and are not directly comparable. Concept shift changes the task more radically, requiring a learner to change its predictions, rather than just become accurate on a wider range of inputs. We conjecture that changes in $P(y|x)$ must therefore be kept smooth enough for the outer loop to have pressure to capitalize on HI-ADS.

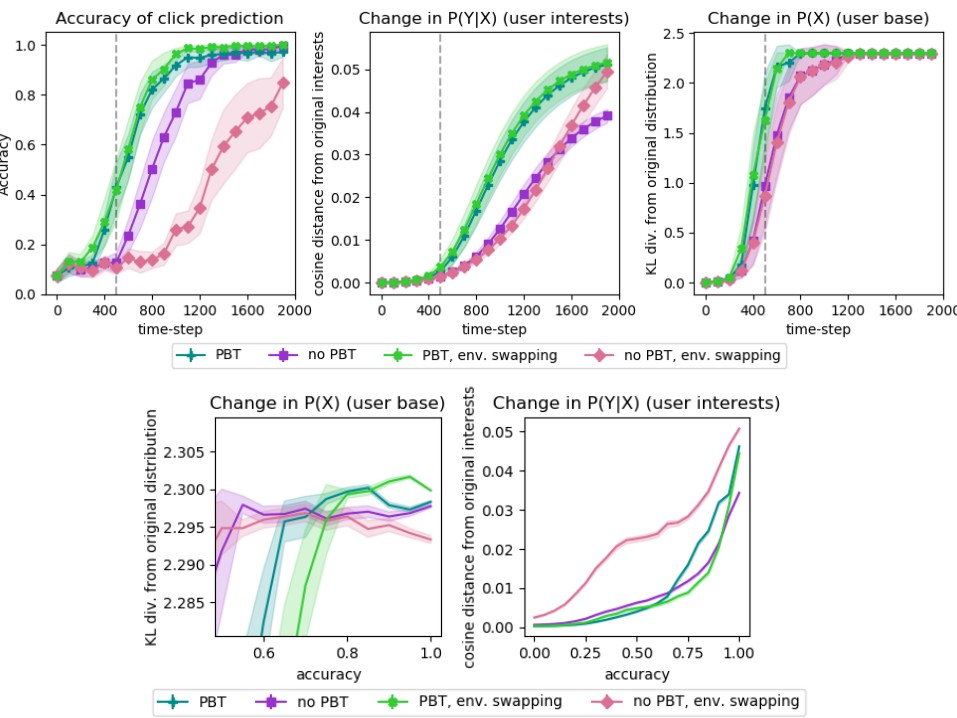

Figure 13: Context swapping doesn't have the desired effect in the content recommendation environment.

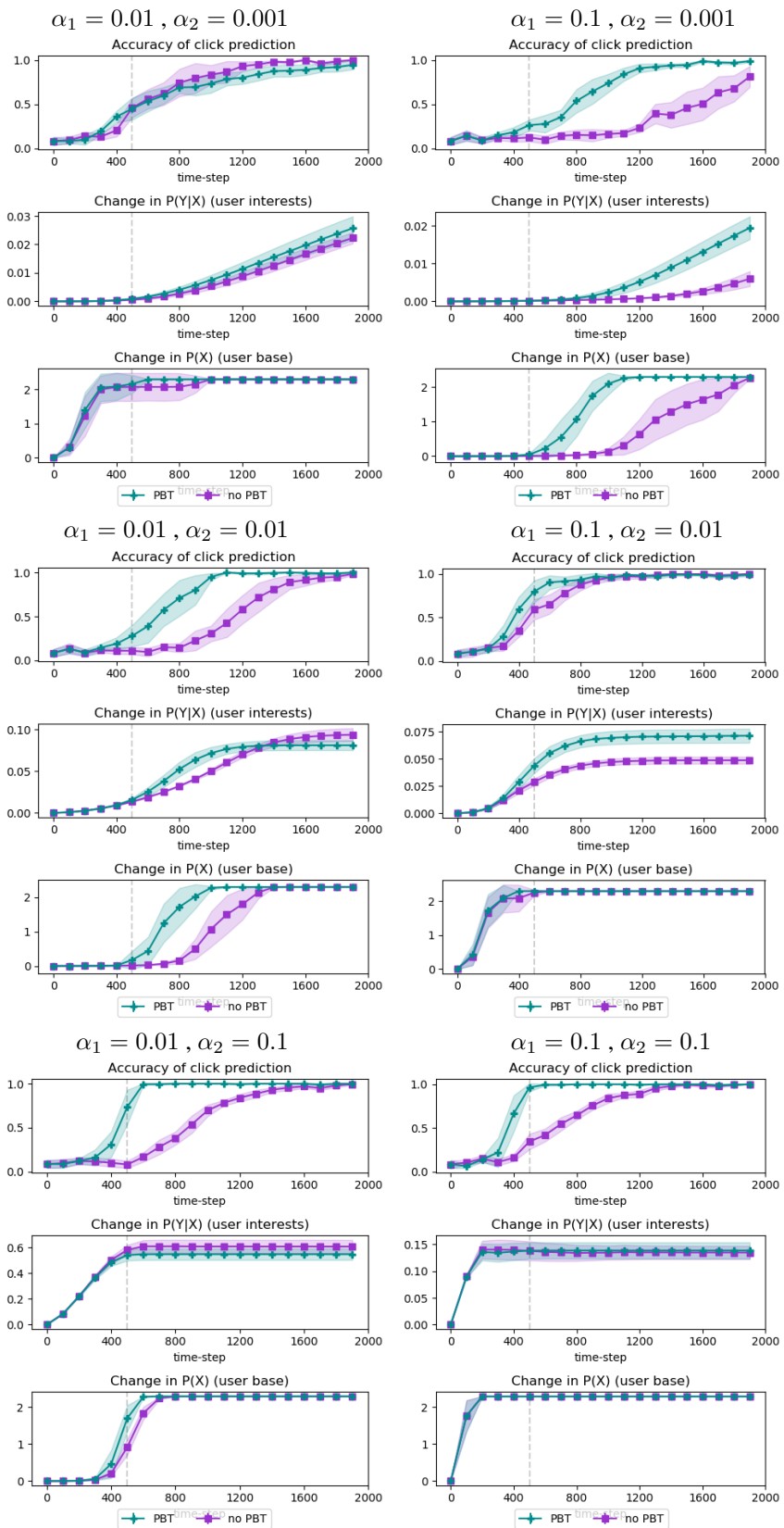

Figure 14: Content recommendation results for different values of $\alpha_1$, $\alpha_2$.

## 10 OFFLINE Q-LEARNING CAN REVEAL INCENTIVES FOR ADS

First, recall that this unit test is a POMDP, and the state is not observed. Since there are only 2 possible actions, a policy is defined by a single parameter $\theta = p(\text{cooperate})$. Now, the state distribution is $P(s = \text{cooperate}) = \theta$ (ignoring the first state, which is appropriate in the limit of infinite data). More specifically, the probability of each state-action combination are as follows:

Suppose we have a dataset of $N$ examples generated by following a fixed policy.

$$Q(C) = \frac{|s = C, a = C|R(s = C, a = C) + |s = D, a = C|R(s = D, a = C)}{|a = C|} \tag{1}$$

$$= \frac{NP(s = C, a = C)R(s = C, a = C) + NP(s = D, a = C)R(s = D, a = C)}{NP(a = C)} \tag{2}$$

$$= \frac{P(s = C, a = C)R(s = C, a = C) + P(s = D, a = C)R(s = D, a = C)}{P(a = C)} \tag{3}$$

$$= \frac{\theta^2 R(s = C, a = C) + \theta(1 - \theta)R(s = D, a = C)}{\theta} \tag{4}$$

$$= \theta R(s = C, a = C) + (1 - \theta)R(s = D, a = C) \tag{5}$$

$$= \theta(0) + (1 - \theta)(-1) \tag{6}$$

$$= \theta - 1 \tag{7}$$

$$Q(D) = \frac{|s = C, a = D|R(s = C, a = D) + |s = D, a = D|R(s = D, a = D)}{|a = D|} \tag{8}$$

$$= \frac{P(s = C, a = D)R(s = C, a = D) + P(s = D, a = D)R(s = D, a = D)}{P(a = D)} \tag{9}$$

$$= P(s = C)R(s = C, a = D) + P(s = D)R(s = D, a = D) \tag{10}$$

$$= \theta(1/2) + (1 - \theta)(-1/2) \tag{11}$$

$$= 1/2(2\theta - 1) \tag{12}$$

$$= \theta - 1/2 \tag{13}$$

So we see that $Q(D) > Q(C)$, regardless of $\theta$.

Now, suppose instead that we have $N$ examples from each of 2 different policies (given by parameters $\theta_1$ and $\theta_2$) operating in different environments. Intuitively, this sort of data might arise in practice from "A/B testing", where 2 different users have been assigned to 2 different policies in order to compare the policies' performance. We now use $DC$ to represent $s = D, a = C$, etc.

$$Q^{\theta_1, \theta_2}(C) = \frac{|CC|R(CC) + |DC|R(DC)}{|C|} \tag{14}$$

$$= \frac{N(P^{\theta_1}(CC) + P^{\theta_2}(CC))R(CC) + N(P^{\theta_1}(DC) + P^{\theta_2}(DC))R(DC)}{N(P^{\theta_1}(C) + P^{\theta_2}(C))} \tag{15}$$

$$= \frac{(P^{\theta_1}(CC) + P^{\theta_2}(CC))R(CC) + (P^{\theta_1}(DC) + P^{\theta_2}(DC))R(DC)}{(P^{\theta_1}(C) + P^{\theta_2}(C))} \tag{16}$$

$$= \frac{(\theta_1^2 + \theta_2^2)R(CC) + (\theta_1(1 - \theta_1) + \theta_2(1 - \theta_2))R(DC)}{\theta_1 + \theta_2} \tag{17}$$

$$= -\frac{\theta_1(1 - \theta_1) + \theta_2(1 - \theta_2)}{\theta_1 + \theta_2} \tag{18}$$

$$= \frac{\theta_1^2 - \theta_1 + \theta_2^2 - \theta_2}{\theta_1 + \theta_2} \tag{19}$$

$$\tag{20}$$

$$Q^{\theta_1,\theta_2}(D) = \frac{|CD|R(CD) + |DD|R(DD)}{|D|} \tag{21}$$

$$= \frac{(P^{\theta_1}(CD) + P^{\theta_2}(CD))R(CD) + (P^{\theta_1}(DD) + P^{\theta_2}(DD))R(DD)}{(P^{\theta_1}(D) + P^{\theta_2}(D))} \tag{22}$$

$$= \frac{1/2(\theta_1(1-\theta_1) + \theta_2(1-\theta_2)) - 1/2((1-\theta_1)^2 + (1-\theta_2)^2)}{2 - \theta_1 - \theta_2} \tag{23}$$

$$= \frac{(2\theta_1 - 1)(1 - \theta_1) + (2\theta_2 - 1)(1 - \theta_2)}{4 - 2\theta_1 - 2\theta_2} \tag{24}$$

Now, in Figure 15 we see that $Q(C) > Q(D)$ when one of the policies cooperates with high probability, and the other defects with high probability. Intuitively, the result of pooling data from 2 such policies is very similar to collecting data from an $\epsilon$-greedy policy trained online (as in Figure 6).

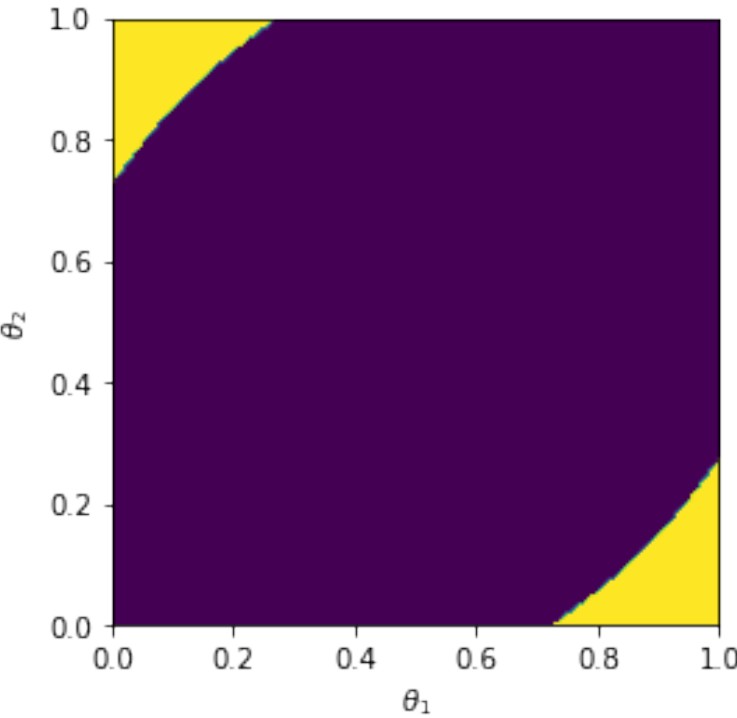

Figure 15: Offline Q-learning can also reveal HI-ADS, when pooling data from different (policy, environment) pairs. Yellow regions represent policy pairs for which $Q(C) > Q(D)$, resulting in non-myopic behavior.

