# OpenReview forum: "Hidden Incentives for Auto-Induced Distributional Shift"
_ICLR.cc/2021/Conference — Reject_

### Official Review · AnonReviewer1 · 2020-10-29
**Official Blind Review #1**

**Rating:** 4
**Confidence:** 3

**Review:**

This paper proposes to study auto-induced distribution shift (ADS), a phenomenon where the machine learning model itself may change its own data distribution via its decisions or actions. The paper seeks to understand, in situations where ADS is undesirable, whether learning algorithms are privy to, and pursue, incentives for learning models which take advantage of ADS.

Pros:

+ The paper proposes an interesting and promising direction for study.
+ Simple examples serve to illustrate the concepts outlined in the paper.

Cons:

- Both expositionally and experimentally, there isn't a compelling argument made in terms of when ADS shows up in real problems.
- There is limited insight as to how the problem of interest may be addressed methodologically.

Weighing these pros and cons, I am inclined to reject this paper. More details are provided below.


Quality
---

Perhaps what detracts from this work the most is the lack of rigorous evaluation on real world problems. It is not difficult to imagine that ADS may indeed be a problem that must be dealt with in certain applications, but it is also unsatisfying to leave this to the imagination. All of the experiments presented are toy, and though toy experiments are certainly valuable for gaining insights into the problem, they should be accompanied also by larger studies that corroborate the story.

The content recommendation experiment seems promising and could perhaps be built out into a larger scale experiment. But this raises additional questions about why the proposed method of context swapping is not effective at handling this situation. In general, the context swapping approach seems rather ad hoc, and might there be potential downsides regarding tradeoffs in model performance? Further investigation into this or other mitigation strategies would also significantly strengthen the work.

Clarity
---

The paper is generally well organized and well written. I appreciate the hierarchy of definitions as well as the discussion of how certain frameworks such as meta learning can exacerbate the problem being studied.

Originality
---

I am not an expert in this area, but the paper seems to propose an original problem to study, and discussion of related work seems relatively well researched and well written.

Significance
---

Relating back to my previous comments, the exposition needs to do a better job at convincing the reader that the problem is important. Further discussion into some of the prior works mentioned in the related work section would help in this regard. Is it possible to characterize prior work into the taxonomy and framework described? Might it even be possible to design (or borrow) an experiment based on this prior work?

In my opinion, the current exposition is not enough to convince the reader that the problem is important. The running content recommendation example seems hypothetical, and a real world system would likely, perhaps even obviously, disincentivize such behavior from a model that drives users away. This indeed may be a hand designed solution, which as the authors correctly point out may not scale to cover all instances of this problem, but it is up to the authors to argue this more convincingly.

---

> ### Author Response · Authors · 2020-11-21
> **Regarding significance**
>
> We appreciate your thoughtful and detailed review.  We’re happy you think our work provides a clear presentation of an “original”, “interesting and promising direction for study”, as this was our main priority.  We aim to address your concerns regarding the significance of this problem and improve the exposition accordingly.  If we succeed in providing a compelling motivation for studying this problem, we hope the contributions of identifying, illustrating, motivating, and exploring an original, interesting, and promising research problem will be sufficient to merit publication.  With this work, we aim to proactively alert the community to this novel issue, in anticipation of its potentially significant technical and social consequences, motivating further study.
>
> **Regarding significance:**
> We believe the main significance of our work is its novelty.  While many previous works have addressed ADS in some way or another, only a handful of papers have been published on the topic of incentives.  Moreover, while previous works have examined which incentives exist in the real world, we are the first to note that incentives can be hidden or revealed, and the consequences for specification.
>
> This is significant because in practice, incentives are routinely hidden, implicitly.  For example, any time we use supervised learning in an online learning context, we don’t expect the supervised learning algorithm to sacrifice accuracy on the current time-step in order to make future examples easier to predict.  Yet doing so would often lead to more accurate predictions.  Typical supervised learning algorithms (e.g. gradient-based training of neural networks) appear to effectively hide these incentives.  But our experiments suggest that other approaches, e.g. evolutionary algorithms such as PBT, may not.  Thus a thorough understanding of how algorithmic choices affect which incentives are revealed or hidden is called for.  While our work emphasized the potential practical importance of this study, we also believe it is of fundamental scientific interest to understand the role of algorithmic choices in hiding or revealing incentives.
>
> Another motivation for our work, which has been widely discussed in the context of AI alignment, is *wireheading* or “reward hijacking”.  For instance, [1] argue that any RL system has an incentive to hijack its reward signal (e.g. gain control of the hardware register where it is stored) and provide itself arbitrarily high reward, rather using the reward as a signal of which behavior is desirable. By understanding and specifying which incentives an algorithm pursues, this kind of behaviour might be prevented.
>
> We’d also like to highlight that driving away users is only *one kind* of problem that can result in the particular application of content recommendation.  We also discuss the problem of changing users’ interests, e.g. leading to radicalization.  Appendix 8 provides a more in-depth discussion of these issues.
> Regarding driving away users: as AnonReviewer2 noted, a system might drive away only user with “complex profiles” (since they are harder to predict), and heuristic approaches might fail to notice and account for such an effect, which might in practice take the form of bias against small minority groups or more unique users (who might also provide more value to the platform).
>
> [1] Cohen, Michael K., Badri N. Vellambi, and Marcus Hutter. "Asymptotically Unambitious Artificial General Intelligence." AAAI. 2020.

---

> ### Author Response · Authors · 2020-11-21
> **Regarding Experiments and Methodology**
>
> **Regarding Experiments:**
> We’re not aware of prior works containing experiments that would be valuable to include in this work.
> Briefly, the prior works we reference do not seem suitable because:
> * Previous works on ADS that we’re aware of either aim to correct for ADS, or to induce ADS in a way that increases performance.
> * Our content recommendation environment is already a slightly *more* complex version of the kind of bandit problem discussed by Shah et al., which is the most relevant previous experiment we could find.
> * Previous work on incentives is either entirely theoretical, or uses toy examples no more sophisticated than ours.
>
> Nonetheless, we are considering possible additional experiments which could be performed during the discussion phase, and appreciate the suggestion to take inspiration from prior work.  However, we also don’t see any clear “hole” in our work, barring a demonstration in a real-world content recommendation setting.  We don’t have access to such a system, and experiments on real users would raise potential ethical issues.
>
> **Regarding methodology for addressing the problem:**
> We believe the correct approach to addressing this problem will likely involve identifying qualitative aspects of learning algorithms that make them more or less likely to reveal incentives.  Our work makes progress in this direction by 1) providing unit tests that make it simple to determine whether a particular algorithm reveals incentives, and 2) providing evidence that meta-learning and “causal confusion” (i.e. using non-causal correlations to guide decision-making) are qualitative features of learning algorithms with the potential to reveal incentives.

---

> > ### Comment · AnonReviewer1 · 2020-11-23
> > **Still concerned about the significance of this work**
> >
> > Thanks to the authors for providing additional comments, they are certainly useful in better placing and understanding this work. And again, this work is quite interesting and novel -- perhaps it is the novelty that makes the work so hard to judge. All of the reviewers, including me, have relatively low confidence scores about our reviews. After reading the authors' comments, I am still not confident that this work is of particular significance in its current form.
> >
> > In fact, in my view, the authors themselves point out several directions that this work may enable but currently does not explore adequately. Perhaps the most interesting direction is in providing a set of tools for determining how different algorithms hide or pursue incentives for ADS, as the authors mention several times:
> >
> > - "Thus a thorough understanding of how algorithmic choices affect which incentives are revealed or hidden is called for."
> > - "we also believe it is of fundamental scientific interest to understand the role of algorithmic choices in hiding or revealing incentives."
> > - "By understanding and specifying which incentives an algorithm pursues, this kind of behaviour might be prevented."
> >
> > It is true that this paper makes steps in this direction, but they appear to be rather small steps, in the form of a couple of unit tests and quantitatively analyzing a couple of algorithms. My primary concern with this is that it may simply be unconvincing in terms of the point that the authors are trying to make. This, in my view, prevents the realization of the authors' goal to "proactively alert the community to this novel issue".

---

> > > ### Author Response · Authors · 2020-11-23
> > > **Significance of unit tests should not be underestimated.  What would you find convincing?**
> > >
> > > Thank you for continuing the discussion!
> > >
> > > Please let us know if there are any specific things we could do (e.g. in terms of experiments) to convince you.
> > >
> > > In your original review, you said
> > > > the exposition needs to do a better job at convincing the reader that the problem is important.
> > >
> > > Our comment titled “regarding significance” aimed to address this concern.
> > > Are you (still) unconvinced that the problem is important?  Or is your concern more about the exposition?
> > > If you are not convinced, can you explain why you found our response unconvincing?  Otherwise, can you elaborate on your concerns about the exposition?
> > >
> > > In contrast, in your most recent comment, it seems like you are suggesting that the lack of significance is *not* because we fail to convince the reader that the problem is important, but rather because our contributions towards solving the problem are “rather small steps” (can you please clarify if we’ve misunderstood?).  We respectfully disagree.  While we acknowledge that much work remains, we believe our unit tests are highly significant for several reasons:
> > > * They provide a *maximally simple* illustration of incentives for ADS; the simplicity is a strength (just like with the original Prisoner’s dilemma!), because it removes unnecessary details.  In our mind, it is hard to imagine experiments that would be more illuminating of the fundamental phenomena of HI-ADS.
> > > * The simplicity is also a strength because it helps facilitate future research.  As we mention, *revealed* incentives are not necessarily *pursued*, but the simplicity of our unit tests means that a learner will almost certainly pursue incentives for ADS if they are revealed.
> > >
> > > We also emphasize again that we think the novelty, potential importance, and subtlety of HI-ADS make clear and compelling presentation of this issue a significant contribution. Our experience discussing this issue with a number of researchers, including previous reviewers, leads us to believe there is already a lot of difficult content to digest in the current version, and any significant expansion of the scope of this work would risk jeopardizing the core objective of our work, which is to provide the reader with a clear understanding of HI-ADS and their potential relevance for specification and alignment.  Indeed, although you found our work “well organized and well written”, we note that AnonReviewer3 still did not find our work entirely clear.  We believe this highlights the challenge (and therefore, significance) of providing a clear presentation of HI-ADS.

---

> > > ### Author Response · Authors · 2020-11-24
> > > **Please see comment to AnonReviewer2 for new results!**
> > >
> > > In our comment titled:
> > > > Examples of problematic incentives for ADS in "batch learning from bandit feedback" settings
> > >
> > > We describe a new result which we believe may help address your concerns of significance.
> > > This result shows that incentives for ADS can be revealed when learning from offline data, which AnonReviewer2 believes to be the "most established setup" for training real-world content recommendation systems.
> > >
> > > We believe this example significantly strengthens our results, and will help make the significance of our work clear and convincing.  Please let us know if you'd like us to elaborate on these results in any way.

---

> > > ### Author Response · Authors · 2020-11-24
> > > **We're not just "quantitatively analyzing a couple of algorithms", we propose and validate 2 *qualitative* properties which lead algorithms to reveal HI-ADS**
> > >
> > > On page 6-7, we explain that our results indicate that these two factors have the potential to reveal HI-ADS:
> > > (1) Optimizing over a longer time-scale (PBT and REINFORCE (as a meta-learning algorithm))
> > > (2) Exploiting correlation (PBT and Q-learning).
> > >
> > > We will emphasize this in our revision, since it does not seem to have been clear enough.

---

### Official Review · AnonReviewer4 · 2020-10-30
**An interesting concept, but not clear in many aspects**

**Rating:** 5
**Confidence:** 2

**Review:**

This paper introduces the concept of auto-induced distributional shift (ADS), and argues that some meta learning and reinforcement learning algorithms have the incentives to change the distribution so that the problem is easier to solve. The paper presents unit tests to detect hidden incentives for auto-induced distributional shift (HI-ADS) and also proposes context swapping to reduce the distributional shift. Experiments on Population-Based Training (PBT) show that PBT reveals HI-ADS in unit tests and context swapping mitigates the distributional shift for PBT.

Pros:
1) The concept of ADS is interesting. Distributional shift is not a new phenomenon especially for reinforcement learning. As mentioned in the paper some existing works try to exploit the distributional shift. And if distributional shift is harmful then algorithm could use a better reward function to mitigate the effects. But this concept is indeed not formally defined and studied in previous works.

2) The results of unit tests reveal hidden incentives of PBT under two toy environments.

I have the following concerns and questions:

1) How to measure HI-ADS is unclear. From the definition of HI-ADS in Section 4.1, it seems that we should evaluate by whether "the learner would not learn to perform the incentivized behaviors at higher than chance levels".  In the unit test, the incentivized behaviors are clearly defined for the two toy environments, but the paper does not discuss how to find / define incentivized behaviors for a general environment. This directly leads to my second concern.

2) How to generalize the unit test? The unit test only tells whether an algorithm reveals ADS in the two toy environments. Even if a meta learning algorithm passes the unit test, it is not guaranteed that such algorithm won't exhibits ADS in other application scenarios. Can we get more from the unit test? In real world applications if we do not have a perfect understanding of the environment that what kind of behaviors are encouraged by incentives, then how to evaluate HI-ADS?

3) It is not clear whether PBT algorithm itself or all meta learning algorithms would exhibit HI-ADS. The authors use PBT as an example to illustrate HI-ADS, but it is unclear whether this observation suggests other meta learning algorithms are also vulnerable to ADS or that we should test them individually using the unit tests.

4) Why would context swapping mitigate the distributional shift? It is not clearly explained. My rough understanding is after switching the environment, the learners are using misspecified models and won't fully exploit current estimation.  Does context swapping help with content recommendation task? Is this a general approach for other environments instead of the toy unit test setting?


Other comments:

1) Figure 5 and 6 are too small to read.

2) Appendix is missing, which is supposed to contain additional details on experiments according to the main content.

---

> ### Author Response · Authors · 2020-11-21
> **Response**
>
> We’re glad you find our work compelling and novel.  We mostly agree with your comments and concerns, but hope to convince you of our work’s value despite these limitations.
>
> **Regarding points 1 and 2 (How to measure HI-ADS in general, and how to generalize the unit tests):**
>
> It is true that we do not have a precise and formal method of defining HI-ADS in general.  This is actually a key motivation for introducing these unit tests.  We believe that the propensity of an algorithm to reveal HI-ADS in the unit tests is indicative of the potential of that same algorithm to reveal HI-ADS in more complicated learning problems.  Our experiments on content recommendation provide support for this hypothesis, since PBT reveals HI-ADS in both the unit tests and the content recommendation environment.  Intuitively, the idea is that the unit tests are as simple as possible, so not revealing HI-ADS in the unit tests should be a strong signal that an algorithm would not reveal them in more challenging settings, where greater capacity might be required to notice the potential to improve performance via ADS.  Thus passing the unit tests strongly suggests, but does not prove, that the algorithm will not reveal HI-ADS in other contexts.
>
> In order to generalize the unit test for HI-ADS to more complicated settings, we could randomly sample a wide variety of learning algorithms that pass the unit tests and use their behavior to define “chance levels” of incentivized behaviors in the complicated environment. Then any algorithm which causes the behaviour at higher-than-chance levels could be considered to fail that test.  However this is likely to be computationally expensive and perhaps impractical for most complicated settings. We believe that a more promising direction for future research is to use these unit tests to generate and evaluate hypotheses about which features of learning algorithms influence their propensity to reveal HI-ADS, with the goal of developing a theoretical understanding of this issue.
>
> >It is not clear whether PBT algorithm itself or all meta learning algorithms would exhibit HI-ADS
>
> We hypothesize that all, or at least most, meta-learning algorithms would exhibit HI-ADS.  Our results with PBT, combined with the results using REINFORCE as a meta-learning algorithm, support this conclusion. Intuitively, meta-learning algorithms -- by design -- optimize over a longer time horizon, observing correlations of past actions/decisions with present performance, and thus revealing an incentive to use those correlations as a means of improving performance.   But since we haven’t proved such a result, or indeed, even formally defined meta-learning, at present we’d recommend applying the unit test when HI-ADS is a concern for *any* algorithm.
>
>
> > why would context swapping mitigate the distributional shift?
>
> Thank you for identifying the lack of clarity on this point. We hypothesize that the reason the outer loop is able to reveal the incentive for ADS is that it ‘sees’ the correlation between its actions and its future performance, and thus learns to favor actions that affect the state of its environment (i.e. use ADS) in a way that increases the future performance.  By swapping the context, the benefits of such actions are instead reaped by another learner, effectively removing the incentive for non-myopic behavior.  Is this clearer?  We would like to improve the discussion of this method.
>
> So we believe the success of context swapping is not due to the learners lack of knowledge of the state (which is how we interpret your statement that “My rough understanding is after switching the environment, the learners are using misspecified models and won't fully exploit current estimation.”), but rather due to removing the correlation between current action and future reward.  We did not find context swapping to help with the content recommendation experiments, see Figure 13 and Section 9.2.4 (Appendix).
>
> > Figure 5 and 6 are too small to read.
>
> Sorry about that, and thanks for mentioning it!  Do you mean the plots themselves, or just the text?  Either way, we’ll fix it in the revision.  You can also see Figure 8(bottom) and Figure 11 in the Appendix for larger versions of Figure 5(B2) and Figure 6.
> Thank you very much for noticing the Appendix was missing, by the way! We very much regret that error!
>
>
> > And if distributional shift is harmful then algorithm could use a better reward function to mitigate the effects.
>
> Can you please clarify this comment?  To be clear, we claim that using a better reward function is not always a promising approach towards addressing ADS (see the first full paragraph on page 2).  Using the content recommendation example, this is because we may not have a clear idea which kinds of changes in user interests are (un)desirable, and so we’d prefer to let user interests evolve “naturally” (i.e. without the AI system exerting any pressure for them to evolve in any particular way).

---

> > ### Author Response · Authors · 2020-11-24
> > **Do you have remaining concerns or questions that we can try to address?**
> >
> > As we understand, your main criticism was a lack of clarity.  Thank you for listing specific questions and concerns!
> > We hope that we've managed to make things clear to you, and will be sure to address these concerns in our rebuttal.
> > Please let us know if you have remaining concerns or questions we can address before the end of the rebuttal period.

---

### Official Review · AnonReviewer2 · 2020-10-30
**Good idea, really interesting problem, but however of limited impact due to its close relationship with Population Based Training**

**Rating:** 6
**Confidence:** 3

**Review:**

Abstract: The paper highlights an interesting problem of learning dynamics, where a learning system has the incentives to change it’s future input in order to increase its performance. This is not always problematic, but it can, at times, lead to perverse incentives for the system, such as under-performing on users with complex profiles in the present, in order to remove them from the future testing sets. The authors propose a set of unit tests to detect this issue and a way to solve the problem for the PBS metalearning frameworks. The experiments on toy/simulated datasets show the relevance of the approach.


Pros:
- Clarity: The paper is quite well-written and the explanations are clear. I think the problem that the paper aims to address is real and of practical value.
- Experimental design: The experimental section, is very clear and supports the claims of the paper

Cons:
- Significance/Impact: I find the scope of the paper to be limited to Population Based Training, which as far as I can tell, its not a well-established training method in real-world applications, especially recommendation. I would like to see the expansion of the same ideas to general batch learning from bandit feedback which is as far as I can tell the most established setup for RW ML decision systems.

---

> ### Author Response · Authors · 2020-11-21
> **Response**
>
> Thank you very much for your review. We are very happy to hear the practical relevance and experiments were clear and well-motivated.
> * The scope of our paper is not limited to PBT. This confusion was shared by other reviewers, and we regret the lack of clarity; we will emphasize and clarify this in our revision.  We demonstrate that REINFORCE (as a meta-learning algorithm) or Q-learning (without meta-learning) can also reveal incentives for ADS, and use these experiments to argue that meta-learning and “causal confusion” (i.e. using non-causal correlations to guide decision-making) are qualitative features of learning algorithms with the potential to reveal incentives.
> * Our understanding is that PBT has been used in production by Google (see our top-level comment: “To all reviewers: Comment about our contributions and the nature of our work.”).
> * Can you please elaborate on what exactly you mean by general batch learning from bandit feedback?

---

> > ### Author Response · Authors · 2020-11-24
> > **Examples of problematic incentives for ADS in "batch learning from bandit feedback" settings**
> >
> > By the “general batch learning from bandit feedback”, we’ll assume that you mean a setting in which data is collected and then learning takes place offline on a fixed dataset.  Please let us know if you had something different, or more specific, in mind.
> >
> > Here we provide an example of how Q-learning could reveal incentives for ADS even in this setting.
> > We will incorporate these results into our revision (and polish the presentation).
> > In the current version of the PDF (Appendix 10), we show that:
> > * If our batch of data is all generated by the same policy, then incentives for ADS are *not* revealed.
> > * If instead, the batch of data contains interactions of 2 different policies in 2 different environments, then incentives for ADS *can* be revealed.
> >
> > This happens when one of these policies cooperates with high probability, and the other defects with high probability.  The result of pooling data from 2 such policies is very similar to pooling data from an $\epsilon$-greedy policy trained online (as in our current Q-learning experiments, Figure 6.
> > As motivation for this scenario, consider a dataset containing data from 2 different users interacting with 2 different content recommendation algorithms (e.g. data collected from “A/B testing”).
> >
> > As another example, we note that a very common form of A/B testing is simply to compare the performance of 2 candidate policies.  In the RL unit test, A/B testing over any extended window of time would favor the non-myopic “(almost) always cooperate” policy over the myopic “(almost) always defect” policy for the same reason that meta-learning does: the average reward of cooperating is higher.  In other words, A/B testing would also fail the unit test.  We will discuss these examples in the revision, and thank you for suggesting these additions.

---

### Official Review · AnonReviewer3 · 2020-11-06
**Interesting idea, but the proposal lacks clarity**

**Rating:** 4
**Confidence:** 3

**Review:**

This paper discusses a phenomenon where machine-learned models may influence user behaviors in future iterations, creating self-selection effects such as filter bubbles or propagation of fake news. The paper calls these effects auto-induced distribution shift (ADS) and argues that a specific meta-learning algorithm PBT manipulates users instead of maximizing rewards. To illustrate the ideas, the paper introduces a few simulation environments where user behaviors may change when certain prediction events happen.

Despite an interesting motivation, the paper is quite vague in the proposed methods. For example,
1. I am not able to associate ADS with the specific meta-learning algorithm called PBT. Self-selection is a universal phenomenon present in many types of algorithms. It can be corrected by any learners that focus on the detection of distribution shifts, e.g., via inverse propensity scoring, or under ignorability assumptions. On the other hand, PBT is just an algorithm with automated hyperparameter search. What can we learn by connecting ADS to PBT?
2. Why are reinforcement learning / supervised learning free from incentive / ADS issues? For the reasons above, I believe these claims are unrelated and wrong.
3. Is learning conducted online or offline? Would user states be observable or unobservable? If offline, how would the learner foresee the shifts in user behaviors after online feedback loop? If online and observable, shouldn't we focus on modeling user state transitions? If online and unobservable, what are the belief states?

Additionally, the experiments are far from being practical. While Experiment 1 makes sense that by sacrificing an immediate reward, the user gets "manipulated" into a state of more predictability, I am not seeing how Experiment 2 makes practical sense. In this example, the environment is set as a modified prisoner's dilemma. The paper assumes "defect" to be a better solution, but in this specific case, it seems that cooperation seems to be the true user utility, despite the user gets "manipulated" by non-myopic RL algorithms.

The paper also lacks general clarity. I am not sure how content swapping works in Section 4.2. The details about Q-learning should be reorganized. Section 5.2 is best presented with equations or diagrams. Overall, the authors should think about how to eliminate the unrelated claims and simplify the paper to include just one proposed method, which solves a wide class of interesting problems at the same time.

---

> ### Author Response · Authors · 2020-11-21
> **Key Clarifications: 1) Tracking distributional shifts does *not* address HI-ADS. 2) This is not a methods paper.  3) Interpretation of prisoner’s dilemma experiment.**
>
> Thank you for your thoughtful review.
> We’d like to clarify some important aspects of our work. Please let us know if any of these remain unclear, or if any were helpful enough to you that you think they should be included in a revision of the manuscript:
> * You’ve correctly characterized ADS.  However, our work is specifically focused on *incentives* for ADS (see Figure 3), which cannot be effectively addressed by “any learners that focus on the detection of distribution shifts”.  The fundamental problem we address in this work is not that the learner fails to track changes in the distribution, but rather that the learner learns to *cause* such changes in order to increase performance.  In the example of content recommendation, this could be undesirable because it corresponds to manipulating the user base.
> * This is not a “methods” paper.  It is an “idea” paper.  Our primary goal is to identify and shed light on issues related to incentives for ADS.  We show how unit tests and context swapping can help manage incentives for ADS, but we do not present these as a complete solution.  But we believe our more significant contribution is to identify factors that can influence whether incentives are visible or hidden, namely: 1) optimizing over a longer time-scale, and 2) exploiting (non-causal) correlations.  This is how PBT is connected to ADS: By optimizing over a longer time horizon than a non-meta learning algorithm, PBT can reveal hidden incentives for ADS, leading to qualitative changes in learner behavior.
> * We believe the prisoner’s dilemma experiment provides a crisp illustration of a central claim: *the desired behavior is not always the one which maximizes the objective function.*  We find this highly counter-intuitive/surprising, and thus significant.
> We indeed assume that defect is the desired behavior!  As justification, consider that a user may want an AI system to obey whatever command it is given, to the best of its ability, regardless of future consequences.  If the AI system knows that obeying that command will make it harder to complete future assignments, and thus decrease future expected reward, then we in fact seek the behavior which produces lower long-term reward!  Note that this is just a more general version of the issue we’ve identified with content recommendation, where a user wants the AI to show them content they are interested in right now, not to manipulate their interests to make them more predictable.

---

> > ### Author Response · Authors · 2020-11-24
> > **On the viability of "detection of distribution shifts" as an approach to HI-ADS: 'not trying to change X' is not the same as 'trying to not change X'**
> >
> > Elaborating on our response to your point (1): Common misconceptions about our work are that incentives for ADS can be addressed by one or both of:
> > * detecting distributional shifts as they occur
> > * preventing distributional shifts from occurring.
> >
> > This additional comment is just meant to emphasize that this is not the case.
> > The important thing to understand is that **"not trying to change X"** (our goal) is not the same as "trying to not change X" or "trying to account for changes in X".  Please let us know if this is/was(n't) clear to you?

---

> ### Author Response · Authors · 2020-11-21
> **Responses to Questions**
>
> * “What can we learn by connecting ADS to PBT?” First, we learn that the learning algorithm plays an important role in specification, since using PBT changes learners’ behavior, *even with no change to the objective function*.  And in these simple unit tests, it is implausible that this is merely an optimization issue; rather, PBT changes the incentivized solutions to the learning problems.  Second, we gain insight into which qualities of PBT lead to this effect, namely: 1) optimizing over a longer time-scale, and 2) exploiting (non-causal) correlations (see bottom pg 6).  Our experiments using REINFORCE (Figure 5-B2) and Q-learning (Figure 6) support our hypothesis that these general principles of PBT are responsible for this effect.
> * “Why are reinforcement learning / supervised learning free from incentive / ADS issues? For the reasons above, I believe these claims are unrelated and wrong.”  The caption of figure 2 where this statement is made is misleading; thank you for identifying this.  Do the smaller captions (a) and (d)  make sense?  What we mean is that neither of these problems (2a, 2d) involves *undesirable incentives* for ADS (which are indicated by orange lines in 2b,2c).  But the reasons for this are different in RL (2a) vs. SL (2d): in (2a), incentives for ADS are desirable; in (2d), ADS does not occur.  We will remove that sentence and more clearly explain the 4 cases depicted.
> * Learning is online and the *type* of the current user is observed, but not their interests; we will clarify this in our revision.  Our model is very simple (an MLP), and does not use belief states; it simply attempts to predict the user’s click.  While modeling user transitions could help a learner manipulate the user base in order to accurately predict clicks, it’s unclear to us how it would help *avoid* manipulation, which is our goal. (see also our “Key clarifications” comment), and please let us know if we misunderstood your suggestion.

---

> ### Author Response · Authors · 2020-11-21
> **Regarding Clarity**
>
> We are happy to include any clarifications you find useful in our revision!
>
> * Context swapping: In Figure 4, A_i represents a specific learner, e.g. a set of parameters for the ML model.  As a concrete example, imagine training a fleet of self-driving cars online.  There are N cars and N different learners.  Context swapping would periodically change which learner is controlling which car. This can remove incentives the learner has to optimize over a longer time horizon in a given environment, because it won’t predictably be in that environment to reap the reward.   Please let us know if this explanation doesn’t clarify this for you.
> * Q-learning: can you offer a more specific suggestion, or let us know what was unclear about the current organization?
> * We include equations in the Appendix (see updated PDF).  Adding a figure is a great idea; we’ll do it.
> * Which “unrelated claims” are you referring to?  We’ve tried to make our paper as focused and clear as possible.  But our focus is to identify and understand problems related to HI-ADS, *not* to propose a method.

---

### Author Response · Authors · 2020-11-18
**Uploaded new version with appendix**

Thanks to AnonReviewer4 for noticing that our original submission was missing the appendix.
We apologize for this oversight, and have uploaded a new version with the appendix included.
Note that the references to sections of the appendix in the original paper were also incorrect, and have been updated.

---

### Author Response · Authors · 2020-11-18
**To all reviewers: Comment about our contributions and the nature of our work.**

We sincerely thank all the reviewers for their work, and will respond to their reviews in individual detail soon.

But first, we believe there may be some subtle, high-level confusion as to how we view the contributions of our work, and we’d like to make sure this is clear (please let us know if not) so that we can clarify it accurately in the manuscript.

Our work is not meant to be about PBT or content recommendation, *per se*.  These are used as illustrative examples. We view our main contributions as clearly identifying and describing the potential problems of ADS and hidden incentives for ADS, and making these problems intuitive and actionable for future research.
The general phenomena we’re interested in are:
* hiding incentives as a method of specifying desired behavior
* ways in which algorithmic design choices (e.g. whether to use a meta-learning algorithm, such as PBT) can influence which incentives are visible/hidden
* how failing to hide/reveal incentives appropriately can lead to undesirable behavior (e.g. user manipulation in content recommendation), and
* how we might begin to address this issue in practice.


Please also note that we do not argue or demonstrate that such failures are causing problems in currently deployed machine learning systems.  We suspect they may be, but note that content recommendation systems are typically proprietary and not subject to public scrutiny, making this difficult to assess.  We believe PBT has been used by Google in production systems; indeed the “DeepMind for Google” team, which focuses on applying DeepMind’s research in Google products, has worked on a more practical variant (see “A Generalized Framework for Population Based Training”, Li et al. 2019); we will note this in our revision.

With this work, we aim to proactively alert the machine learning community to this novel issue, in anticipation of its potentially significant technical and social consequences, and to motivate further study.
We’ve found this topic challenging to explain well in the past.  Among other reasons, we think this is due to (1) the counter-intuitive notion that we might prefer lower-performance behavior from an algorithm, as long as the algorithm is solving the task as intended / not causing undesirable side-effects; (2) the subtlety of the distinction between a system that “tries to prevent ADS” and one that “doesn’t try to induce ADS” (the latter being our focus); and (3) the interdisciplinary nature of the problem, touching on aspects of social science, game theory, and social science.
Thus our focus is simply to clearly communicate the relevant concepts, and provide some simple tools (the unit tests) for researchers interested in this topic.
We hope this helps to clarify the modest scope of our intended contribution, and that reviewers will find this issue important enough to merit acceptance if we are able to address individual concerns.

---

> ### Author Response · Authors · 2020-11-20
> **Addendum to previous comment: when content recommender systems use reinforcement learning, incentives are revealed *by design***
>
> Regarding the question of whether incentives for user manipulation are a problem in real-world content recommender systems:
> Using (non-myopic) reinforcement learning for content recommendation clearly has the potential to reveal incentives for user manipulation.  Both Facebook and Youtube are reported to use or have used reinforcement learning:
> https://research.fb.com/wp-content/uploads/2018/10/Horizon-Facebooks-Open-Source-Applied-Reinforcement-Learning-Platform.pdf
> https://www.newamerica.org/oti/reports/why-am-i-seeing-this/case-study-youtube/
>
> The extent to which designers are attempting to avoid/hide incentives for user manipulation at present is unclear, but we expect that there will be increasing efforts made in this direction.  Our work is predicated on the assumption that designers are making such an effort to hide these incentives, e.g. by using myopic reinforcement learning, or supervised learning.  Our findings (with PBT/meta-learning and Q-learning) show how such efforts might unexpectedly fail.
>
> Again, ours is forward-looking work; we want these problems to be well-known and understood before they become critical.

---

### Decision · Program_Chairs · 2021-01-07
**Final Decision**

**Decision:**

Reject

**Comment:**

This is a thought-provoking paper which describes a significant problem that plausibly occurs in deployed ML/RL models.
The paper is clearly written, describing claims using examples and developing small unit-tests to probe models.
However, as the reviews and discussion show,  the exposition should be substantially re-worked so that the core contributions are more understandable -- the core message in the revised manuscript is still very nuanced and easy to mis-understand.

Let's say we train an ML model using supervised learning to minimize a loss function on a dataset. Several models may have near-optimal loss as measured on a validation set -- a learning algorithm is free to return any one of them. Now in a deployed system, these ML models are not merely generating passive predictions; these predictions are driving system operation and potentially influencing future states/contexts/inputs that the model will be invoked on. It is well known that supervised learning makes an iid assumption between training and deployment which is violated in this setting -- that is not the main point of this paper. Consider again the set of models with near-optimal loss. Some of them, when deployed, may cause the distribution mismatch between training and deployment to be miniscule, while other models may introduce a vast mismatch. We may choose a learning algorithm which just so happens to pick models from the former category; and we may conclude that feedback effects induced by the ML model are not substantial. We then change some unrelated detail in the learning algorithm (but not the objective, datasets, validation criteria, etc.) which just so happens to pick models from the latter category and suddenly witness a large distribution shift. What happened? And could we have developed tests to detect that our learning algorithms have these tendencies? The paper attempts to articulate such questions, and design the first step in answering them.

Moving to RL, where we routinely consider distribution shifts in states visited by different policies, does not fundamentally fix all these issues because the reward function is typically an engineered proxy to elicit desired behavior -- and we may again find that some RL algorithms have a tendency to find reward-maximizing policies that exploit gaps in reward specification as opposed to following intended behavior.

The core question studied in this paper, scoped to the supervised learning setting, is very related to that of strategic classification (see e.g., https://arxiv.org/pdf/1910.10362.pdf Strategic Classification is Causal Modeling in Disguise). The following sketch is inspired by that literature.

We might hope to augment the training objective of ML/myopic RL/strategic RL to address the Auto-induced distribution shift problem as follows.
[Supervised learning for content recommendation] Let the training/validation data distribution be D. Assume for now that there is no exogenous factor in the environment that causes any distribution shifts in deployment -- so, the only shift is due to feedback effects from the predictions made by the model. For an ML model f, let the corresponding recommendation policy be pi_f, and let the long-term distribution of data seen from user-interactions with pi_f be D[pi_f]. Then, what we want is: f* = argmin_F Expectation over D [ L(f) ] subject to constraint that D ~= D[pi_f].
For a contextual bandit/myopic RL formulation of the problem, we could similarly constrain the learning problem as pi* = argmax_Pi Expectation over D [ Reward of pi] subject to constraint that D \approx D[pi].
Essentially, both supervised ML and contextual bandit algorithms are assuming that context distribution is unchanged -- so let us enforce that the context distribution is indeed unchanged as a consequence of the policy's actions.
It is unclear how to generalize this kind of thinking to situations where environmental changes also contribute to distribution shift.  The authors call out precisely this flaw using the cryptic comment -- 'not trying to change X' is not the same as 'trying to not change X'. The formulation above does 'trying to not change X', but that is an insufficient band-aid in situations when environment changes X. It's also unclear how one might estimate D[pi] or D[pi_f] and appropriately constrain the learning algorithm -- but these are all interesting questions to study.

The paper in its current form is asking an important question. In supervised learning, the desired solution might actually coincide with strategic classification solution concepts. The paper may be asking a generalization of the phenomenon for myopic RL and RL. It may spark interesting discussions and follow-up work, but is not yet mature beyond a workshop poster.
Generalizing the unit-tests, articulating the scope of situations where context-swapping may be a useful strategy, and even formalizing the problem and desired goal (as attempted above for the content recommendation example) will substantially strengthen the paper.